# FiGURe: Simple and Efficient Unsupervised Node Representations with Filter Augmentations

**Chanakya Ekbote**[*]
Microsoft Research India
chanakyaekbote@gmail.com

**Ajinkya Pankaj Deshpande***
Microsoft Research India
ajinkya.deshpande56@gmail.com

**Arun Iyer**
Microsoft Research India
ariy@microsoft.com

**Ramakrishna Bairi**
Microsoft Research India
rkbairi@gmail.com

**Sundararajan Sellamanickam**
Microsoft Research India
ssrajan@microsoft.com

## Abstract

Unsupervised node representations learnt using contrastive learning-based methods have shown good performance on downstream tasks. However, these methods rely on augmentations that mimic low-pass filters, limiting their performance on tasks requiring different eigen-spectrum parts. This paper presents a simple filter-based augmentation method to capture different parts of the eigen-spectrum. We show significant improvements using these augmentations. Further, we show that sharing the same weights across these different filter augmentations is possible, reducing the computational load. In addition, previous works have shown that good performance on downstream tasks requires high dimensional representations. Working with high dimensions increases the computations, especially when multiple augmentations are involved. We mitigate this problem and recover good performance through lower dimensional embeddings using simple random Fourier feature projections. Our method, FiGURe, achieves an average gain of up to 4.4%, compared to the state-of-the-art unsupervised models, across all datasets in consideration, both homophilic and heterophilic. Our code can be found at: https://github.com/microsoft/figure.

## 1 Introduction

Contrastive learning is a powerful method for unsupervised graph representation learning, achieving notable success in various applications [44, 15]. However, these evaluations typically focus on tasks exhibiting homophily, where task labels strongly correlate with the graph's structure. An existing edge suggests the connected nodes likely share similar labels in these scenarios. However, these representations often struggle when dealing with heterophilic tasks, where edges tend to connect nodes with different labels. Several papers [7, 17, 4, 28] have tackled the problem of heterophily by leveraging information from both low and high-frequency components. However, these methods operate in the semi-supervised setting, and the extension of these ideas in unsupervised learning still needs to be explored. Inspired by the insights in these papers, we propose a simple method incorporating these principles.

Our approach introduces filter banks as additional views and learns separate representations for each filter bank. However, this approach faces two main challenges: Firstly, storing representations from each view can become prohibitively expensive for large graphs; secondly, contrastive learning

---

[*]Both authors contributed equally to this work. Work done while the authors were Research Fellows at Microsoft Research India.

methods typically demand high-dimensional representations, which increase both the computational cost of training and the storage burden. We employ a shared encoder for all filter banks to tackle the first challenge. Our results confirm that a shared encoder performs on par with independent encoders for each filter bank. This strategy enables us to reconstruct filter-specific representations as needed, drastically reducing the storage requirement. For the second challenge, we train our models with low-dimensional embeddings. Then, we use random Fourier feature projection [40] to lift these low-dimensional embeddings into a higher-dimensional space. Kernel tricks [23] were typically used in classical machine learning to project low-dimensional representation to high dimensions where the labels can become linearly separable. However, constructing and leveraging the kernels in large dataset scenarios could be expensive. To avoid this issue, several papers [40, 41, 20, 36, 26] proposed to approximate the map associated with the kernel. For our scenario, we use the map associated with Gaussian kernel [40]. We empirically demonstrate that using such a simple approach preserves high performance for downstream tasks, even in the contrastive learning setting. Consequently, our solution offers a more efficient approach to unsupervised graph representation learning in computation and storage, especially concerning heterophilic tasks. The proposed method exhibits simplicity not only in the augmentation of filters but also in its ability to learn and capture information in a low-dimensional space, while still benefiting from the advantages of large-dimensional embeddings through Random Fourier Feature projections.

Our contributions in this work are, 1] We propose a simple scheme of using filter banks for learning representations that can cater to both heterophily and homophily tasks, 2] We address the computational and storage burden associated with this simple strategy by sharing the encoder across these various filter views, 3] By learning a low-dimensional representation and later projecting it to high dimensions using random Fourier Features, we further reduce the burden, 4] We study the performance of our approach on four homophilic and seven heterophilic datasets. Our method, FiGURe, achieves an average gain of up to 4.4%, compared to the state-of-the-art unsupervised models, across all datasets in consideration, both homophilic and heterophilic. Notably, even without access to task-specific labels, FiGURe performs competitively with supervised methods like GCN [25].

## 2  Related Work

Several unsupervised representation learning methods have been proposed in prior literature. Random walk-based methods like Node2Vec [14] and DeepWalk [37] preserve node proximity but tend to neglect structural information and node features. Contrastive methods, such as DEEP GRAPH INFOMAX (DGI) [44], maximize the mutual information (MI) between local and global representations while minimizing the MI between corrupted representations. Methods like MVGRL [15] and GRACE [45] expand on this, by integrating additional views into the MI maximization objective. However, most of these methods focus on the low frequency components, overlooking critical insights from other parts. Semi-supervised methods like GPRGNN [7], BERNNET [17], and PPGNN [28] address this by exploring the entire eigenspectrum, but these concepts are yet to be applied in the unsupervised domain. This work proposes the use of a filter bank to capture information across the full eigenspectrum while sharing an encoder across filters. Given the high-dimensional representation demand of contrastive learning methods, we propose using Random Fourier Features (RFF) to project lower-dimensional embeddings into higher-dimensional spaces, reducing computational load without sacrificing performance. The ensuing sections define our problem, describe filter banks and random feature maps, and explain our model and experimental results.

## 3  Problem Setting

In the domain of unsupervised representation learning, our focus lies on graph data, denoted as $\mathcal{G} = (\mathcal{V}, \mathcal{E})$, where $\mathcal{V}$ is the set of vertices and $\mathcal{E}$ the set of edges ($\mathcal{E} \subseteq \mathcal{V} \times \mathcal{V}$). We associate an adjacency matrix with $\mathcal{G}$, referred to as $\mathbf{A} : \mathbf{A} \in \{0, 1\}^{n \times n}$, where $n = |\mathcal{V}|$ corresponds to the number of nodes. Let $\mathbf{X} \in \mathbb{R}^{n \times d}$ be the feature matrix. We use $\mathbf{A_I}$ to represent $\mathbf{A} + \mathbf{I}$ with $\mathbf{I}$ is the identity matrix, while $\mathbf{D_{A_I}}$ signifies the degree matrix of $\mathbf{A_I}$. We also define $\mathbf{A_n}$ as $\mathbf{D_{A_I}}^{-1/2} \mathbf{A_I} \mathbf{D_{A_I}}^{-1/2}$. No additional information is provided during training. The goal is to learn a parameterized encoder, $E_\theta : \mathbb{R}^{n \times n} \times \mathbb{R}^{n \times d} \mapsto \mathbb{R}^{n \times d'}$, where $d' \ll d$. This encoder produces a set of node representations $E_\theta(\mathbf{X}, \mathbf{A_n}) = \{h_1, h_2, ..., h_n\}$ where each $h_i \in \mathbb{R}^{d'}$ represents a rich representation for node $i$. The

subsequent section will provide preliminary details about filter banks and random feature maps before we discuss the specifics of the proposed approach.

## 4 Preliminaries

Our approach relies on filter banks and random feature maps. In this section, we briefly introduce these components, paving the way for a detailed explanation of our approach.

### 4.1 Filter Banks

Graph Fourier Transform (GFT) forms the basis of Graph Neural Networks (GNNs). A GFT is defined using a reference operator $\mathbf{R}$ which admits a spectral decomposition. Traditionally, in the case of GNNs, this reference operator has been the symmetric normalized laplacian $\mathbf{L_n} = \mathbf{I} - \mathbf{A_n}$ or the $\mathbf{A_n}$ as simplified in [25]. A graph filter is an operator that acts independently on the entire eigenspace of a diagonalisable and symmetric reference operator $\mathbf{R}$, by modulating their corresponding eigenvalues. [43, 42]. Thus, a graph filter $\mathbf{H}$ is defined via the graph filter function $g(.)$ operating on the reference operator as $\mathbf{H} = g(\mathbf{R}) = \mathbf{U}g(\mathbf{\Lambda})\mathbf{U}^T$. Here, $\mathbf{\Lambda} = diag([\lambda_1, \lambda_2, ..., \lambda_n])$, where $\lambda_i$ denotes the eigenvalues of the reference operator. We describe a filter bank as a set of filters, denoted as $\mathbf{F} = \{\mathbf{F}_1, \mathbf{F}_2, ..., \mathbf{F}_K\}$. Both GPRGNN [7] and BERNNET [17] employ filter banks, comprising of polynomial filters, and amalgamate the representations from each filter bank to enhance the performance across heterophilic datasets. GPRGNN uses a filter bank defined as $\mathbf{F}_{\text{GPRGNN}} = \{\mathbf{I}, \mathbf{A_n}, ..., \mathbf{A_n}^{K-1}\}$, while $\mathbf{F}_{\text{BERNNET}} = \{\mathbf{B}_0, \mathbf{B}_1, ..., \mathbf{B}_{K-1}\}$ characterizes the filter bank utilized by BERNNET. Here, $\mathbf{B}_i = \frac{1}{2^{K-1}}\binom{K-1}{i}(2\mathbf{I} - \mathbf{L_n})^{K-i-1}(\mathbf{L_n})^i$. Each filter in these banks highlights different parts of the eigenspectrum. By tuning the combination on downstream tasks, it offers the choice to select and leverage the right spectrum to enhance performance. Notably, unlike traditional GNNs, which primarily emphasize low-frequency components, higher frequency components have proved useful for heterophily [4, 7, 17, 28]. Consequently, a vital takeaway is that **for comprehensive representations, we must aggregate information from different parts of the eigenspectrum and fine-tune it for specific downstream tasks**.

### 4.2 Random Feature Maps for Kernel Approximations

Before the emergence of deep learning models, the kernel trick was instrumental in learning non-linear models. A kernel function, $k : \mathbb{R}^d \times \mathbb{R}^d \mapsto \mathbb{R}$, accepts two input features and returns a real-valued score. Given a positive-definite kernel, Mercer's Theorem [30] assures the existence of a feature map $\phi(\cdot)$, such that $k(x, y) = \langle \phi(x), \phi(y) \rangle$. Leveraging the kernel trick, researchers combined Mercer's theorem with the representer theorem [23], enabling the construction of non-linear models that remain linear in $k$. These models created directly using $k$ instead of the potentially complex $\phi$, outperformed traditional linear models. The implicit maps linked with these kernels projected the features into a significantly high-dimensional space, where targets were presumed to be linearly separable. However, computational challenges arose when dealing with large datasets. Addressing these issues, subsequent works [41, 20, 36, 40] introduced approximations of the map associated with individual kernels through random projections into higher-dimensional spaces ($\phi'(.)$). This approach ensures that $\langle \phi'(\mathbf{x}), \phi'(\mathbf{y}) \rangle \approx k(x, y)$. These random feature maps are inexpensive to compute and affirm that simple projections to higher-dimensional spaces can achieve linear separability. The critical insight is that **computationally efficient random feature maps, such as Random Fourier features (RFF) [40], exist. These maps project lower-dimensional representations into higher dimensions, enhancing their adaptability for downstream tasks.**.

## 5 Proposed Approach

The following section delineates the process of unsupervised representation learning. Then, we detail the use of filter bank representations in downstream tasks with random feature maps.

### 5.1 Unsupervised Representation Learning

Our method FiGURe (**Fi**lter-based **G**raph **U**nsupervised **Re**presentation Learning) builds on concepts introduced in [18, 44], extending the maximization of mutual information between node and global

filter representations for each filter in the filter bank $\mathbf{F} = \{\mathbf{F}_1, \mathbf{F}_2, ...\mathbf{F}_K\}$. In this approach, we employ filter-based augmentations, treating filter banks as "additional views" within the context of contrastive learning schemes. In the traditional approach, alternative baselines like DGI have employed a single filter in the GPRGNN filter bank. Additionally, MVGRL attempted to use the diffusion kernel; nevertheless, they only learned a single representation per node. We believe that this approach is insufficient for accommodating a wide range of downstream tasks. We construct an encoder for each filter to maximize the mutual information between the input data and encoder output. For the $i^{\text{th}}$ filter, we learn an encoder, $E_\theta : \mathcal{X}_i \to \mathcal{X}_i'$, denoted by learnable parameters $\theta$. In this context, $\mathcal{X}_i$ represents a set of examples, where each example $[\widehat{\mathbf{X}_{ij}}, \widehat{\mathbf{F}_{ij}}] \in \mathcal{X}_i$ consists of a filter $\mathbf{F}_i$, its corresponding nodes and node features, drawn from an empirical probability distribution $\mathbb{P}_i$. That is, $\mathbb{P}_i$ captures the joint distribution of a filter $\mathbf{F}_i$, its corresponding nodes and node features ($[\mathbf{X}, \mathbf{F}_i]$). Note that $[\widehat{\mathbf{X}_{ij}}, \widehat{\mathbf{F}_{ij}}]$ denote nodes, node features and edges sampled from the $i^{th}$ filter ($[\mathbf{X}, \mathbf{F}_i]$) basis the probability distribution $\mathbb{P}_i$. Note that $\widehat{\mathbf{X}_{ij}} \in \mathbb{R}^{N' \times d}$ and $\widehat{\mathbf{F}_{ij}} \in \mathbb{R}^{N' \times N'}$. $\mathcal{X}_i'$ defines the set of representations learnt by the encoder on utilizing feature information as well as topological information from the samples, sampled from the joint distribution $\mathbb{P}_i$. The goal, aligned with [29, 18, 44], is to identify $\theta$ that maximizes mutual information between $[\mathbf{X}, \mathbf{F}_i]$ and $E_\theta(\mathbf{X}, \mathbf{F}_i)$, or $\mathcal{I}_i([\mathbf{X}, \mathbf{F}_i], E_\theta(\mathbf{X}, \mathbf{F}_i))$. While exact mutual information (MI) computation is infeasible due to unavailable exact data and learned representations distributions, we can estimate the MI using the Jensen-Shannon MI estimator [9, 32], defined as:

$$\mathcal{I}_{i,\theta,\omega}^{\text{JSD}}([\mathbf{X}, \mathbf{F}_i], E_\theta(\mathbf{X}, \mathbf{F}_i)) := \mathbb{E}_{\mathbb{P}_i}[-\text{sp}(-T_{\theta,\omega}([\widehat{\mathbf{X}_{ij}}, \widehat{\mathbf{F}_{ij}}], E_\theta(\widehat{\mathbf{X}_{ij}}, \widehat{\mathbf{F}_{ij}})))] - \\ \mathbb{E}_{\mathbb{P}_i \times \widetilde{\mathbb{P}}_i}[\text{sp}(T_{\theta,\omega}([\widehat{\mathbf{X}_{ij}}, \widehat{\mathbf{F}_{ij}}], E_\theta[\widetilde{\mathbf{X}_{ij}}, \widetilde{\mathbf{F}_{ij}}]))] \quad (1)$$

Here, $T_\omega : \mathcal{X}_i \times \mathcal{X}_i' \to \mathbb{R}$ represents a discriminator function with learnable parameters $\omega$. Note that $[\widetilde{\mathbf{X}_{ij}}, \widetilde{\mathbf{F}_{ij}}]$ is an input sampled from $\widetilde{\mathbb{P}}_i$, which denotes a distribution over the corrupted input data (more details given below). The function sp(.) corresponds to the softplus function [10]. Additionally, $T_{\theta,\omega}([h_{ij}]_1, [h_{ij}]_2) = D_w \circ (\mathcal{R}([h_{ij}]_1), [h_{ij}]_2)$, where $\mathcal{R}$ denotes the readout function responsible for summarizing all node representations by aggregating and distilling information into a global filter representation. We introduce a learnable discriminator $D_\omega$, where $D_\omega(.,.)$ represents the joint probability score between the global representation and the node-specific patch representation. Note that $[h_{ij}]_1$, denotes representations obtained after passing samples sampled from the original distribution $\mathbb{P}_i$, to the encoder, and $[h_{ij}]_2$, denotes representations obtained after passing samples sampled from the original distribution $\mathbb{P}_i$ or samples sampled from the corrupted distribution $\widetilde{\mathbb{P}}_i$ to the encoder. Intuitively, Eq. 1 implies that we would want to maximize the mutual information between the local (patch) representations of the nodes and their corresponding global graph

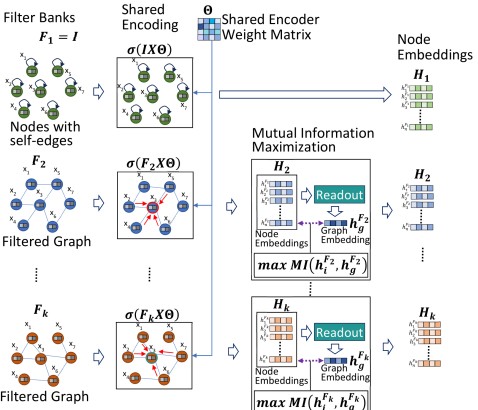

Figure 1: Unsupervised learning of node embeddings by maximizing mutual information between node and graph representations over the graphs from the filter bank. Note that the parameter $\Theta$ is shared across all the filters.

representation, while minimising the mutual information between a global graph representation and the local representations of corrupted input data. Note that: $[\widetilde{\mathbf{X}_{ij}}, \widetilde{\mathbf{F}_{ij}}]$ denotes a corrupted version of input features and given filter. More details with regards to this will be given below. In our approach, we first obtain node representations by feeding the filter-specific topology and associated node features into the encoder: $\mathbf{H}_{ij} = E_\theta(\mathbf{X}_{ij}, \mathbf{F}_{ij}) = \{h_1^{\mathbf{F}_{ij}}, h_2^{\mathbf{F}_{ij}}, ..., h_{N'}^{\mathbf{F}_{ij}}\}$. Note that $\mathbf{H}_{ij}$ has the following dimensions $\mathbb{R}^{N' \times d'}$. To obtain global representations, we employ a readout function $\mathcal{R} : \mathbb{R}^{N' \times d'} \to \mathbb{R}^{d'}$, which combines and distills information into a global representation $h_g^{F_{ij}} = \mathcal{R}(\mathbf{H}_{ij}) = \mathcal{R}(E_\theta(\mathbf{X}_{ij}, \mathbf{F}_{ij}))$. Instead of directly maximizing the mutual information between the local and global representations, we maximize $D_\omega(.,.)$. This joint score should be

higher when considering global and local representations obtained from the same filter, as opposed to the joint score between the global representation from one filter and the local representation from corrupted $(\mathbf{X}_{ij}, \mathbf{F}_{ij})$. To generate negative samples for contrastive learning, we employ a corruption function $\mathcal{C} : \mathbb{R}^{N' \times d} \times \mathbb{R}^{N' \times N'} \to \mathbb{R}^{N' \times d} \times \mathbb{R}^{N' \times N'}$, which yields corrupted samples denoted as $[\widetilde{\mathbf{X}_{ij}}, \widetilde{\mathbf{F}_{ij}}] = \mathcal{C}(\mathbf{X}_{ij}, \mathbf{F}_{ij})$. The designed corruption function generates data decorrelated with the input data. Note that $[\widetilde{\mathbf{X}_{ij}}, \widetilde{\mathbf{F}_{ij}}]$ denote nodes, node features and edges sampled from the corrupted version of $i^{th}$ filter, basis the probability distribution $\widetilde{\mathbb{P}}_i$. The corruption function can be designed basis the task at hand. We employ a simple corruption function, whose details are present in the experimental section. Let the corrupted node representations be as follows: $\widetilde{\mathbf{H}}_{ij} = E_\theta(\mathcal{C}(\mathbf{X}_{ij}, \mathbf{F}_{ij})) = E_\theta(\widetilde{\mathbf{X}_{ij}}, \widetilde{\mathbf{F}_{ij}}) = \{h_1^{\widetilde{\mathbf{F}_{ij}}}, h_2^{\widetilde{\mathbf{F}_{ij}}}, ..., h_{N'}^{\widetilde{\mathbf{F}_{ij}}}\}$. In order to learn representations across all filters in the filter bank, we aim to maximise the average estimate of mutual information (MI) across all filters, considering $K$ filters, defined by $\mathcal{I}_{\mathbf{F}}$.

$$\mathcal{I}_{\mathbf{F}} = \frac{1}{K} \sum_{i=1}^{K} \mathcal{I}_{i,\theta,\omega}^{JSD}([\mathbf{X}, \mathbf{F}_i], E_\theta(\mathbf{X}, \mathbf{F}_i)) \tag{2}$$

Maximising the Jenson-Shannon MI estimator can be approximately optimized by reducing the binary cross entropy loss defined between positive samples (sampled from the $\mathbb{P}_i$) and the negative samples (sampled from $\widetilde{\mathbb{P}}_i$). Therefore, for each filter, the loss for is defined as follows:

$$\mathcal{L}_{\mathbf{F}_i 1} = -\frac{1}{2N'} \mathbb{E}_{([\mathbf{X}_{ij}, \mathbf{F}_{ij}] \sim \mathbb{P}_i)} \left( \sum_{k=1}^{N'} [\log(D_\omega(h_k^{\mathbf{F}_{ij}}, h_g^{\mathbf{F}_{ij}})) + \log(1 - D_\omega(h_k^{\widetilde{\mathbf{F}_{ij}}}, h_g^{\mathbf{F}_{ij}}))] \right) \tag{3}$$

Note that for Eq. 3, the global representation $(h_g^{\mathbf{F}_{ij}})$ is generated by $\mathcal{R}(E_\theta(\mathbf{X}_{ij}, \mathbf{F}_{ij}))$. The local representations $(h_k^{\widetilde{\mathbf{F}_{ij}}} \forall k)$ are constructed by passing the sampled graph and features through a corruption function (see $\widetilde{\mathbf{H}}_{ij}$). Therefore to learn meaningful representations across all filters the following objective is minimised:

$$\mathcal{L} = \frac{1}{K} \sum_{i=1}^{K} \mathcal{L}_{\mathbf{F}_i} \tag{4}$$

Managing the computational cost and storage demands for large graphs with distinct node representations for each filter poses a challenge, especially when contrastive learning methods require high dimensions. To address this, we employ parameter sharing, inspired by studies like [7] and [17]. This approach involves sharing the encoder's parameters $\theta$ and the discriminator's parameters $\omega$ across all filters. Instead of storing dense filter-specific node representations, we only store the shared encoder's parameters and the first-hop neighborhood information for each node per filter, reducing storage requirements. To obtain embeddings for downstream tasks, we reconstruct filter-specific representations using a simple one-layer GNN. This on-demand reconstruction significantly reduces computational and storage needs associated with individual node representations. Fig 1 illustrates such a simple encoder's mutual information-based learning process. To address the second challenge, we initially train our models to produce low-dimensional embeddings that capture latent classes, as discussed in [2]. These embeddings, while informative, lack linear separability. To enhance separability, we project these low-dimensional embeddings into a higher-dimensional space using random Fourier feature (RFF) projections, inspired by kernel methods (see Section 4.2). This approach improves the linear separability of latent classes, as confirmed by our experimental results in Section 6.2, demonstrating the retention of latent class information in these embeddings.

### 5.2 Supervised Representation Learning

After obtaining representations for each filter post the reconstruction of the node representations, learning an aggregation mechanism to combine information from representations that capture different parts of the eigenspectrum for the given task is necessary. We follow learning schemes from [7, 17, 28],

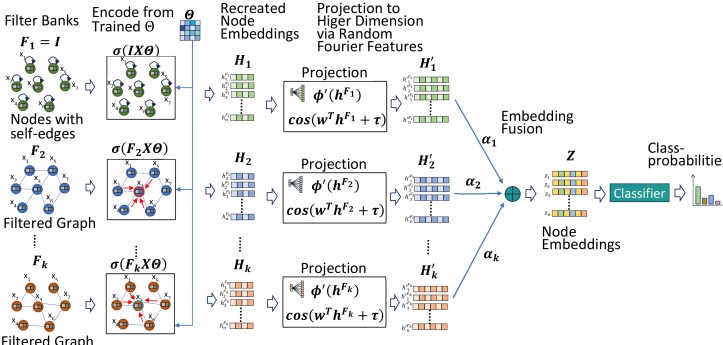

Figure 2: Supervised Learning: Using the trained parameter $\Theta$, we generate the node embeddings by encoding the filtered graphs that get consumed in the classification task.

learning a weighted combination of filter-specific representations. The combined representations for downstream tasks, considering $K$ filters from filter bank $\mathbf{F}$, are as follows:

$$Z = \sum_{i=1}^{K} \alpha_i \phi'(E_\theta(\mathbf{X}, \mathbf{F}_i)) \tag{5}$$

The parameters $\alpha_i$'s are learnable. Additionally, the function $\phi'(.)$ represents either the RFF projection or an identity transformation, depending on whether $E_\theta(\mathbf{X}, \mathbf{F}_i)$ is low-dimensional or not. A classifier model (e.g. logistic regression) consumes these embeddings, where we train both the $\alpha_i$'s and the weights of the classifier. Fig 2 illustrates this process. Notably, semi-supervised methods like [28, 7, 17] differ as they learn both encoder and coefficients from labeled data, while our method pre-trains the encoder and learns task-specific combinations of filter-specific representations.

## 6 Experimental Results

**Training Details**: We define a single-layer graph convolutional network (GCN) with shared weights ($\Theta$) across all filters in the filter bank ($\mathbf{F}$) as our encoder. Therefore, the encoder can be expressed as follows: $E_\theta(\mathbf{X}, \mathbf{F}_i) = \psi(\mathbf{F}_i\mathbf{X}\Theta)$. It is important to note that $\mathbf{F}_i$ represents a normalized filter with self-loops, which ensures that its eigenvalues are within the range of [0, 2]. The non-linearity function $\psi$ refers to the parametric rectified linear unit [16]. As we work with a single graph, we obtain the positive samples by sampling nodes from the graph. Using these sampled nodes, we construct a new adjacency list that only includes the edges between these sampled nodes in filter $\mathbf{F}_i$. On the other hand, the corruption function $\mathcal{C}$ operates on the same sampled nodes. However, it randomly shuffles the node features instead of perturbing the adjacency list. In essence, this involves permuting the node features while maintaining a consistent graph structure. This action introduces a form of corruption, altering each node's feature to differ from its original representation in the data. Similar to [44], we employ a straightforward readout function that involves averaging the representations across all nodes for a specific filter $\mathbf{F}_i$: $\mathcal{R}(\mathbf{H}_i) = \sigma\left(\frac{1}{N}\sum_{j=0}^{N} h_j^{\mathbf{F}_i}\right)$ where $\sigma$ denotes the sigmoid non-linearity. We utilize a bilinear scoring function $D_\omega(.,.)$, whose parameters are also shared across all filters, where $D_\omega(h_j^{\mathbf{F}_i}, h_g^{\mathbf{F}_i}) = \sigma(h_j^{\mathbf{F}_i T}\mathbf{W} h_g^{\mathbf{F}_i})$. We learn the encoder and discriminator parameters by optimising Eq. 4. While we could use various filter banks, we specifically employ the filter bank corresponding to GPRGNN ($\mathbf{F}_{\text{GPRGNN}}$) for all our experiments. However, we conduct an ablation study (see 6.5) comparing $\mathbf{F}_{\text{GPRGNN}}$ with $\mathbf{F}_{\text{BERNNET}}$. Additional training details are available in 8.3.

We conducted a series of comprehensive experiments to evaluate the effectiveness and competitiveness of our proposed model compared to SOTA models and methods. These experiments address the following research questions: **[RQ1]** How does FiGURe, perform compared to SOTA unsupervised models? **[RQ2]** Can we perform satisfactorily even with lower dimensional representations using projections such as RFF? **[RQ3]** Does shared encoder decrease performance? **[RQ4]** What is the computational efficiency gained by using lower dimensional representations compared to methods that rely on higher dimensional representations? **[RQ5]** Can alternative filter banks be employed to

recover good quality representations? **[RQ6]** How does FiGURe combine information from different filters (in $\mathbf{F}_{\text{GPRGNN}}$) in a task-dependent manner?

**Datasets and Setup:** We evaluated our model on a diverse set of real-world datasets, which include both heterophilic and homophilic networks, to assess its effectiveness. Similar to previous works, we utilized the node classification task as a proxy to evaluate the quality of the learned representations. Please refer to 8.2 for detailed information about the benchmark datasets.

**Baselines:** In our comparison against baselines, we considered common unsupervised approaches, such as DEEPWALK and NODE2VEC, and state-of-the-art mutual information-based methods, namely DGI, MVGRL, GRACE, and SUGRL. We also include the performance numbers of the widely used GCN for reference. Unless stated otherwise, we use a 512-dimensional representation size for all reported results, following prior research. For additional details please refer to 8.3 and 8.4.

## 6.1 RQ1: FiGURe versus SOTA Methods

Table 1: Contains node classification accuracy percentages on homophilic and heterophilic datasets. $\text{FiGURe}_{32}^{\text{RFF}}$ and $\text{FiGURe}_{128}^{\text{RFF}}$ refer to FiGURe trained with 32 and 128 dimensional representations, respectively, and then projected using RFF. Other models, including $\text{FiGURe}_{512}$ and all baselines are trained at 512 dimensions and do not utilize RFF. Higher numbers indicate better performance. It is worth noting that $\text{FiGURe}_{512}$ outperforms or remains competitive with the baselines in all cases. The rightmost column Av. $\Delta_{gain}$ represents the average accuracy % gain of $\text{FiGURe}_{512}$ over the model in that row, averaged across the different datasets. If a model is labeled as 'OOM' for a dataset, it's excluded from Av. $\Delta_{gain}$ calculation. Magenta, Green and Blue represent the 1st, 2nd and 3rd best performing models, for a particular dataset.

| | HETEROPHILIC DATASETS | | | | | HOMOPHILIC DATASETS | | | | |
|---|---|---|---|---|---|---|---|---|---|---|
| | SQUIRREL | CHAMELEON | ROMAN-EMPIRE | MINESWEEPER | ARXIV-YEAR | CORA | CITESEER | PUBMED | OGBN-ARXIV | Av. $\Delta_{gain}$ |
| DEEPWALK | 38.66 (1.44) | 53.42 (1.73) | 13.08 (0.59) | 79.96 (0.08) | 41.05 (0.10) | 83.64 (1.85) | 63.66 (3.36) | 80.85 (0.44) | 64.02 | 13.48 |
| NODE2VEC | 42.60 (1.15) | 54.23 (2.30) | 12.12 (0.30) | 80.00 (0.00) | 39.69 (0.09) | 78.19 (1.14) | 57.45 (6.44) | 73.24 (0.59) | 60.20 | 15.78 |
| DGI | 39.61 (1.81) | 59.28 (1.23) | 47.54 (0.76) | 82.51 (0.47) | 40.59 (0.09) | 84.57 (1.22) | 73.96 (1.61) | 86.57 (0.52) | 65.58 | 6.61 |
| MVGRL | 39.90 (1.39) | 54.61 (2.29) | 68.50 (0.38) | 85.60 (0.35) | OOM | 86.22 (1.30) | 75.02 (1.72) | 87.12 (0.35) | OOM | 4.39 |
| GRACE | 53.15 (1.10) | 68.25 (1.77) | 47.83 (0.53) | 80.22 (0.45) | OOM | 84.79 (1.51) | 67.60 (2.01) | 87.04 (0.43) | OOM | 5.54 |
| SUGRL | 43.13 (1.36) | 58.60 (2.04) | 39.40 (0.49) | 82.40 (0.58) | 36.96 (0.19) | 81.21 (2.07) | 67.50 (1.62) | 86.90 (0.54) | 65.80 | 8.64 |
| $\text{FiGURe}_{32}^{\text{RFF}}$ | 48.89 (1.55) | 65.66 (2.52) | 64.61 (0.92) | 85.28 (0.71) | 41.30 (0.21) | 82.56 (0.87) | 71.25 (2.20) | 83.91 (0.69) | 66.58 | 3.65 |
| $\text{FiGURe}_{128}^{\text{RFF}}$ | 48.78 (2.48) | 66.03 (2.19) | 67.01 (0.56) | 85.16 (0.58) | 41.94 (0.15) | 86.14 (1.13) | 73.34 (1.91) | 83.56 (0.34) | 69.11 | 2.53 |
| $\text{FiGURe}_{512}$ | 52.23 (1.19) | 68.55 (1.87) | 70.99 (0.52) | 85.58 (0.49) | 42.26 (0.20) | 87.00 (1.24) | 74.77 (2.00) | 88.60 (0.44) | 69.69 | 0.00 |

Table 2: Comparing node classification with the widely-used GCN, $\text{FiGURe}_{512}$ achieves strong results even without task-specific labels, highlighting its ability to learn quality representations.

| | SQUIRREL | CHAMELEON | ROMAN-EMPIRE | MINESWEEPER | ARXIV-YEAR | CORA | CITESEER | PUBMED | OGBN-ARXIV |
|---|---|---|---|---|---|---|---|---|---|
| GCN | 47.78 (2.13) | 61.43 (2.70) | **73.69 (0.74)** | **89.75 (0.52)** | **46.02 (0.26)** | **87.36 (0.91)** | **76.47 (1.34)** | 88.41 (0.46) | 69.37 (0.00) |
| $\text{FiGURe}_{512}$ | **52.23 (1.19)** | **68.55 (1.87)** | 70.99 (0.52) | 85.58 (0.49) | 42.26 (0.20) | 87.00 (1.24) | 74.77 (2.00) | **88.60 (0.44)** | **69.69 (0.00)** |

Table 3: Mean epoch time (in seconds) on the two large datasets for different embedding sizes. For lower dimensional embeddings, there is a significant speedup.

| | 512 dims | 128 dims | 32 dims |
|---|---|---|---|
| ARXIV-YEAR | 1.24s | 0.75s | 0.72s |
| OGBN-ARXIV | 0.92s | 0.74s | 0.72s |

We analyzed the results in Table 1 and made important observations. Across homophilic and heterophilic datasets, $\text{FiGURe}_{512}$ consistently outperforms several SOTA unsupervised models, except in a few cases where it achieves comparable performance. Even on the large-scale datasets ARXIV-YEAR and OGBN-ARXIV $\text{FiGURe}_{512}$ performs well, demonstrating the scalability of our method. Two baseline methods MVGRL and GRACE run into memory issues on the larger datasets and are accordingly reported OOM in the table. We want to emphasize the rightmost column of the table, which shows the average percentage gain in performance across all datasets. This metric compares the improvement that $\text{FiGURe}_{512}$ provides over each baseline model for each dataset and averages these improvements. This metric highlights that $\text{FiGURe}_{512}$ performs consistently well across diverse datasets. No other baseline model achieves the same consistent performance across all datasets as $\text{FiGURe}_{512}$. Even the recent state-of-the-art contrastive models GRACE and SUGRL experience

average performance drops of approximately 5% and 10%, respectively. This result indicates that FiGURe$_{512}$ learns representations that exhibit high generalization and task-agnostic capabilities. Another important observation is the effectiveness of RFF projections in improving lower dimensional representations. We compared FiGURe at different dimensions, including FiGURe$_{32}^{RFF}$ and FiGURe$_{128}^{RFF}$, corresponding to learning 32 and 128-dimensional embeddings, respectively, in addition to the baseline representation size of 512 dimensions. Remarkably, even at lower dimensions, FiGURe with RFF projections demonstrates competitive performance across datasets, surpassing the 512-dimensional baselines in several cases. This result highlights the effectiveness of RFFprojections in enhancing the quality of lower dimensional representations. Using lower-dimensional embeddings reduces the computation time and makes FiGURe faster than the baselines. The computational efficiency of reducing dimension size becomes more significant with larger datasets, as evidenced in Table 3. On ARXIV-YEAR, a large graph with 169, 343 nodes, 128-dimensional embeddings yield a 1.6x speedup, and 32-dimensional embeddings yield a 1.7x speedup. Similar results are observed in OGBN-ARXIV. For further insights into the effectiveness of RFFprojections, see Section 6.2, and for computational efficiency gains, refer to Section 6.4. In Table 2, we include GCN as a benchmark for comparison. Remarkably, FiGURe$_{512}$ remains competitive across most datasets, sometimes even surpassing GCN. This highlights that FiGURe$_{512}$ can capture task-specific information required by downstream tasks, typically handled by GCN, through unsupervised means. When considering a downstream task, using FiGURe$_{512}$ embeddings allows for the use of computationally efficient models like Logistic Regression, as opposed to training resource-intensive end-to-end graph neural networks. There are, however, works such as [6], [13], and [5], that explore methods to speed up end-to-end graph neural network training. In summary, FiGURe offers significant computational efficiency advantages over end-to-end supervised graph neural networks. Performance can potentially improve further by incorporating non-linear models like MLP. For detailed comparisons with other supervised methods, please refer to 8.4. It's noteworthy that both OGBN-ARXIV and ARXIV-YEAR use the ARXIV citation network but differ in label prediction tasks (subject area and publication year, respectively). FiGURe demonstrates improvements in both cases, showcasing its task-agnostic and multi-task capabilities due to its flexible node representation. This flexibility enables diverse tasks to extract the most relevant information (see Section 5.2), resulting in strong overall performance. Note that in 8.5, we have also performed an ablation study where the depth of the encoder is increased.

## 6.2 RQ2: RFF Projections on Lower Dimensional Representations

Table 4: Node classification accuracy percentages with and without using Random Fourier Feature projections (on 32 dimensions). A higher number means better performance. The performance is improved by using RFF in almost all cases, indicating the usefulness of this transformation

|  | RFF | CORA | CITESEER | SQUIRREL | CHAMELEON |
|---|---|---|---|---|---|
| DGI | × | **81.65 (1.90)** | 65.62 (2.39) | 31.60 (2.19) | 45.48 (3.02) |
|  | ✓ | 81.49 (1.96) | **66.50 (2.44)** | **38.19 (1.52)** | **56.01 (2.66)** |
| MVGRL | × | **81.03 (1.29)** | 72.38 (1.68) | 37.20 (1.22) | 49.65 (2.08) |
|  | ✓ | 80.48 (1.71) | **72.54 (1.89)** | **39.53 (1.04)** | **56.73 (2.52)** |
| SUGRL | × | 65.35 (2.41) | 42.84 (2.57) | 31.62 (1.47) | 43.20 (1.79) |
|  | ✓ | **70.06 (1.24)** | **47.03 (3.02)** | **38.50 (2.19)** | **51.01 (2.26)** |
| GRACE | × | 76.84 (1.09) | 58.40 (3.05) | 38.20 (1.38) | 53.25 (1.58) |
|  | ✓ | **79.15 (1.44)** | **63.66 (2.96)** | **51.56 (1.39)** | **67.39 (2.23)** |
| FiGURe$_{32}$ | × | **82.88 (1.42)** | 70.32 (1.98) | 39.38 (1.35) | 53.27 (2.40) |
|  | ✓ | 82.56 (0.87) | **71.25 (2.20)** | **48.89 (1.55)** | **65.66 (2.52)** |

In this section, we analyse the performance of unsupervised baselines using 32-dimensional embeddings with and without RFF projections (see Table 4). Despite extensive hyperparameter tuning, we could not replicate the results reported by SUGRL, so we present the best results we obtained. Two noteworthy observations emerge from these tables. Firstly, it is evident that lower dimensional embeddings can yield meaningful and linearly separable representations when combined with simple RFF projections. Utilising RFF projections enhances performance in almost all cases, highlighting the value captured by MI-based methods even with lower-dimensional embeddings. Secondly, FiGURe$_{32}^{RFF}$ consistently achieves superior or comparable performance to the baselines, even in lower dimensions. Notably, this includes SUGRL, purported to excel in such settings. However, there is a 2-

3% performance gap between GRACE and our method for the SQUIRREL and CHAMELEON datasets. While GRACE handles heterophily well at lower dimensions, its performance deteriorates with homophilic graphs, unlike FiGURe$_{32}^{\text{RFF}}$ which captures lower frequency information effectively. Additionally, our method exhibits computational efficiency advantages for specific datasets in lower dimensions. Please refer to 8.6: for discussions with regards to the RFF algorithm, 8.7: for analysing the RFF behaviour and community structure, 8.8: for experiments using other random projection methods, 8.9 and 8.10: for ablation studies with regards to projecting to higher dimensional spaces via RFF, and 8.11: for issues related to including RFF in training. Overall, these findings highlight the potential of RFF projections in extracting useful information from lower dimensional embeddings and reaffirm the competitiveness of FiGURe over the baselines.

## 6.3 RQ3: Sharing Weights Across Filter Specific Encoders

Table 5: A comparison of the performance on the downstream node classification task using independently trained encoders and weight sharing across encoders is shown. The reported metric is accuracy. In both cases, the embeddings are combined using the method described in 5.2

|  | CORA | CITESEER | SQUIRREL | CHAMELEON |
| --- | --- | --- | --- | --- |
| INDEPENDENT | 86.92 (1.10) % | 75.03 (1.75) % | 50.52 (1.51) % | 66.86 (1.85) % |
| SHARED | 87.00 (1.24) % | 74.77 (2.00) % | 52.23 (1.19) % | 68.55 (1.87) % |

Our method proposes to reduce the computational load by sharing the encoder weights across all filters. It stands to reason whether sharing these weights causes any degradation in performance. We present the results with shared and independent encoders across the filters in Table 5 to verify this. We hypothesize that, sharing encoder weights embeds diverse filter representations in a common space, improving suitability for combined representation learning. This enhances features for downstream tasks, in some cases boosting performance. Experimental results confirm that shared weights do not significantly reduce performance; sometimes, they even enhance it, highlighting shared encoders' effectiveness in reducing computational load without sacrificing performance.

## 6.4 RQ4: Computational Efficiency

Table 6: Mean epoch time (in milliseconds) averaged across 20 trials with different hyperparameters. A lower number means the method is faster. Even though our method is slower at 512 dimensions, using 128 and 32 dimensional embeddings significantly reduces the mean epoch time. Using RFF as described in 6.2 we are able to prevent the performance drops experienced by DGI and MVGRL.

|  | DGI | MVGRL | FiGURe$_{512}$ | FiGURe$_{128}^{\text{RFF}}$ | FiGURe$_{32}^{\text{RFF}}$ |
| --- | --- | --- | --- | --- | --- |
| CORA | 38.53 (0.77) | 75.29 (0.56) | 114.38 (0.51) | 20.10 (0.46) | 11.54 (0.34) |
| CITESEER | 52.98 (1.15) | 102.41 (0.99) | 156.24 (0.56) | 30.30 (0.60) | 17.16 (0.51) |
| SQUIRREL | 87.06 (2.07) | 168.24 (2.08) | 257.65 (0.76) | 47.72 (1.40) | 23.52 (1.14) |
| CHAMELEON | 33.08 (0.49) | 64.71 (1.05) | 98.36 (0.64) | 18.56 (0.39) | 11.63 (0.48) |

To assess the computational efficiency of the different methods, we analyzed the computation time and summarized the results in Table 6. The key metric used in this analysis is the mean epoch time: the average time taken to complete one epoch of training. We compared our method with other MI based methods such as DGI and MVGRL. Due to the increase in the number of augmentation views, there is an expected increase in computation time from DGI to MVGRL to FiGURe. However, as demonstrated in 6.2, using RFF projections allows us to achieve competitive performance even at lower dimensions. Therefore, we also included comparisons with our method at 128 and 32 dimensions in the table. It is evident from the results that our method, both at 128 and 32 dimensions, exhibits faster computation times compared to both DGI and MVGRL, which rely on higher-dimensional representations to achieve good performance. This result indicates that FiGURe is computationally efficient due to its ability to work with lower-dimensional representations. During training, our method, FiGURe$_{32}^{\text{RFF}}$, is $\sim$ 3x faster than DGI and $\sim$ 6x times faster than MVGRL. Despite the faster computation, FiGURe$_{32}^{\text{RFF}}$ also exhibits an average performance improvement of around 2% across the datasets over all methods considered in our experiments. Please refer to 8.12 and 8.13 for further discussions.

## 6.5 RQ5: Experiments on Other Filter Banks

Table 7: Accuracy percentages for various filter banks in conjunction with FiGURe. Specifically, $\mathbf{F}^3_{\text{BERNNET}}$ and $\mathbf{F}^{11}_{\text{BERNNET}}$ refer to the $\mathbf{F}_{\text{BERNNET}}$ filter bank with $K$ set to 3 and 11, respectively. Similarly, $\mathbf{F}^3_{\text{CHEBNET}}$ and $\mathbf{F}^{11}_{\text{CHEBNET}}$ represent the $\mathbf{F}_{\text{CHEBNET}}$ filter bank with $K$ set to 3 and 11, respectively.

|  | CORA | CITESEER | SQUIRREL | CHAMELEON |
|---|---|---|---|---|
| $\mathbf{F}^3_{\text{BERNNET}}$ | 85.13 (1.26) | 73.38 (1.81) | 37.07 (1.29) | 53.95 (2.78) |
| $\mathbf{F}^{11}_{\text{BERNNET}}$ | 86.62 (1.59) | 73.97 (1.43) | 43.48 (3.80) | 62.13 (3.66) |
| $\mathbf{F}^3_{\text{CHEBNET}}$ | 83.84 (1.36) | 71.92 (2.29) | 40.23 (1.58) | 60.61 (2.03) |
| $\mathbf{F}^{11}_{\text{CHEBNET}}$ | 76.14 (6.80) | 59.89 (8.94) | 52.46 (1.10) | 67.37 (1.60) |
| $\mathbf{F}_{\text{GPRGNN}}$ | 87.00 (1.24) | 74.77 (2.00) | 52.23 (1.19) | 68.55 (1.87) |

To showcase the versatility of our proposed framework, we conducted an experiment using Bernstein and Chebyshev filters, as detailed in Table 7. The results indicate that using $\mathbf{F}_{\text{GPRGNN}}$ leads to better performance than BERNNET and CHEBNET filters. We believe this is happening is due to the latent characteristics of the dataset. [17, 28] have shown that datasets like CHAMELEON and SQUIRREL need frequency response functions that give more prominence to the tail-end spectrum. GPRGNN filters are more amenable to these needs, as demonstrated in [28]. However, different datasets may require other frequency response shapes, where BERNNET and CHEBNET filters may excel, and give better performance. For instance, $\mathbf{F}_{\text{BERNNET}}$ may better approximate comb filters, as their basis gives uniform prominence to the entire spectrum. Our framework is designed to accommodate any filter bank, catering to diverse dataset needs. Further discussions are in 8.14.

## 6.6 RQ6: Combining information from different filters (in $\mathbf{F}_{\text{GPRGNN}}$)

To analyze how FiGURe combines representations from different filters, we present alpha coefficients for the highest-performing split in Table 8, utilizing $\mathbf{F}_{\text{GPRGNN}}$ on: CORA, CITESEER, SQUIRREL, and CHAMELEON (coefficients may vary for different splits within the same dataset) [28]. GPRGNN filters adapt well to heterophilic datasets like CHAMELEON and SQUIRREL, emphasizing spectral tails with significant weightage on the $\mathbf{A}^2$ filter. In contrast, homophilic datasets require low-pass filters, achieved by assigning higher weightage to the $\mathbf{A}$ and $\mathbf{A}^3$ filters. Achieving these filter shapes with other methods like BERNNET and CHEBNET is possible but more challenging for the model to learn. We believe that this is why GPRGNN filters consistently outperform other filter banks on the datasets we've examined. However, it's important to note that some datasets may benefit from different response shapes, where BERNNET and CHEBNET filters might be more suitable. This table validates our hypothesis about the efficacy of GPRGNN filter coefficients, with $\mathbf{A}^3$ dominating in homophilic datasets (CORA , CITESEER) and $\mathbf{A}^2$ in heterophilic datasets (CHAMELEON , SQUIRREL).

Table 8: We present the alpha coefficients obtained from the best-performing split utilizing GPRGNN filters for CORA, CITESEER, SQUIRREL, and CHAMELEON datasets

|  | I | $\mathbf{A}$ | $\mathbf{A}^2$ | $\mathbf{A}^3$ |
|---|---|---|---|---|
| CORA | 18.2 | 0 | 0 | 35.95 |
| CITESEER | 0 | 0 | 0 | 0.48 |
| SQUIRREL | 0 | 0 | 15.3 | 0 |
| CHAMELEON | 0 | 0 | 8.93 | 0.1 |

## 7  Conclusion and Future Work

Our work demonstrates the benefits of enhancing contrastive learning methods with filter views and learning filter-specific representations to cater to diverse tasks from homophily to heterophily. We have effectively alleviated computational and storage burdens by sharing the encoder across these filters and focusing on low-dimensional embeddings that utilize high-dimensional projections, a technique inspired by random feature maps developed for kernel approximations. Future directions involve expanding the analysis from [2] to graph contrastive learning and investigating linear separability in lower dimensions, which could strengthen the connection to the random feature maps approach.

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

# 8 Supplementary Material

## Contents

## 8.1 Reproducibility

We strive to ensure the reproducibility of our research findings. To facilitate this, we provide the details of our experimental setup, including dataset sources, preprocessing steps, hyperparameters, and model configurations. We also make our code and the datasets used, publicly available at https://github.com/microsoft/figure, enabling researchers to reproduce our results and build upon our work. We would like to emphasize that our code is built on top of the existing MVGRL codebase. For the datasets used in our evaluation, we provide references to their original sources and any specific data splits that we employed. This allows others to obtain the same datasets and perform their own analyses using consistent data. Additionally, we specify the versions of libraries and frameworks used in our experiments, in Section 8.3, and in the REQUIREMENTS file and the README file, in the codebase, enabling others to set up a compatible environment. We document any specific seed values or randomization procedures that may affect the results. By providing these details and resources, we aim to promote transparency and reproducibility in scientific research. We encourage fellow researchers to reach out to us if they have any questions or need further clarification on our methods or results.

## 8.2 Datasets

**Homophilic Datasets:** We evaluated our model (as well as baselines) on three homophilic datasets: CORA, CITESEER, and PUBMED as borrowed from [34]. All three are citation networks, where each node represents a research paper and the links represent citations. Pubmed consists of medical research papers. The task is to predict the category of the research paper. We follow the same dataset setup mentioned in [34] to create 10 random splits for each of these datasets.

**Heterophilic Datasets:** In our evaluation, we included four heterophilic datasets: CHAMELEON, SQUIRREL, ROMAN-EMPIRE, and MINESWEEPER. For CHAMELEON and SQUIRREL, nodes represent Wikipedia web pages and edges capture mutual links between pages. We utilized the ten random splits provided in [34], where 48%, 32%, and 20% of the nodes were allocated for the train, validation, and test sets, respectively. In ROMAN-EMPIRE each node corresponds to a word in the Roman Empire Wikipedia article. Two words are connected with an edge if either these words follow each other in the text, or they are connected in the dependency tree of the sentence. The syntactic role of the word/node defines its class label. The MINESWEEPER graph is a regular 100x100 grid where each node is connected to eight neighboring nodes, and the features are on-hot encoded representations of the number of neighboring mines. The task is to predict which nodes are mines. For both ROMAN-EMPIRE and MINESWEEPER, we used the ten random splits provided in [38].

**Large Datasets:** We also evaluate our method on two large datasets OGBN-ARXIV (from [19]) and ARXIV-YEAR (from [27]). Both these datasets are from the arxiv citation network. In OGBN-ARXIV, the task is to predict the category of the research paper, and in ARXIV-YEAR the task is to predict the year of publication. We use the publicly available splits for OGBN-ARXIV [22] and follow the same dataset setup mentioned in [27] to generate 5 random splits for ARXIV-YEAR. Note that OGBN-ARXIV is a homophilic dataset while ARXIV-YEAR is a heterophilic datasets.

The detailed dataset statistics can be found in Table 9.

Table 9: Dataset Statistics. The table provides information on the following dataset characteristics: number of nodes, number of edges, feature dimension, number of classes, as well as the count of nodes used for training, validation, and testing.

| PROPERTIES | HETEROPHILIC DATASETS | | | | | HOMOPHILIC DATASETS | | | |
|---|---|---|---|---|---|---|---|---|---|
| | SQUIRREL | CHAMELEON | ROMAN-EMPIRE | MINESWEEPER | ARXIV-YEAR | OGBN-ARXIV | CITESEER | PUBMED | CORA |
| #NODES | 5201 | 2277 | 22662 | 10000 | 169343 | 169343 | 3327 | 19717 | 2708 |
| #EDGES | 222134 | 38328 | 32927 | 39402 | 1166243 | 1335586 | 12431 | 108365 | 13264 |
| #FEATURES | 2089 | 500 | 300 | 7 | 128 | 128 | 3703 | 500 | 1433 |
| #CLASSES | 5 | 5 | 18 | 2 | 5 | 40 | 6 | 3 | 7 |
| #TRAIN | 2496 | 1092 | 11331 | 5000 | 84671 | 90941 | 1596 | 9463 | 1192 |
| #VAL | 1664 | 729 | 5665 | 2500 | 42335 | 29799 | 1065 | 6310 | 796 |
| #TEST | 1041 | 456 | 5666 | 2500 | 42337 | 48603 | 666 | 3944 | 497 |

## 8.3 Training Details

We conducted all experiments on a machine equipped with an Intel(R) Xeon(R) CPU E5-2690 v4 @ 2.60GHz processor, 440GB RAM, and a Tesla-P100 GPU with 16GB of memory. The experiments were executed using Python 3.9.12 and PyTorch 1.13.0 [33]. To optimize the hyperparameter search, we employed Optuna [1]. We utilized the Adam optimizer [24] for the optimization process.

### 8.3.1 Unsupervised Training

We conducted hyperparameter tuning for all unsupervised methods using 20 Optuna trials. The hyperparameter ranges and settings for each method are as follows:

DEEPWALK: We set the learning rate to $0.01$, number of epochs to $20$ and the varied the random walk length over $\{8, 9, 10, 11, 12\}$. Additionally, we varied the context window size over $\{3, 4, 5\}$ and the negative size (number of negative samples per positive sample) over $\{4, 5, 6\}$.

NODE2VEC: For Node2Vec, we set the learning rate to $0.01$ and number of epochs to $100$. We varied the number of walks over $\{5, 10, 15\}$ and the walk length over $\{40, 50, 60\}$. The $p$ (return parameter) value was chosen from $\{0.1, 0.25, 0.5, 1\}$ and $q$ (in-out parameter) value was chosen from $\{3, 4, 5\}$.

DGI: DGI [44] proposes a self-supervised learning framework for graph representation learning by maximizing the mutual information between local and global structural context of nodes, enabling unsupervised feature extraction in graph neural networks. We relied on the authors' code[2] and the prescribed hyperparameter ranges specific to the DGI model, for our experiments.

MVGRL: MVGRL [15] proposes a method for learning unsupervised node representations by leveraging two views of the graph data, the graph diffusion view and adjacency graph view. We relied on the authors' code[3] and the prescribed hyperparameter ranges specific to the MVGRL model, for our experiments.

GRACE: GRACE [45] proposes a technique where two different perspectives of the graph are created through corruption, and the learning process involves maximizing the consistency between the node representations obtained from these two views. We relied on the authors' code[4] and the prescribed hyperparameter ranges specific to the GRACE model, for our experiments.

SUGRL: SUGRL [31] proposes a technique for learning unsupervised representations which capture node proximity, while also utilising node feature information. We relied on the authors' code[5] and the prescribed hyperparameter ranges specific to the SUGRL model, for our experiments.

FiGURe: We followed the setting of the MVGRL model, setting the batch size to 2 and number of GCN layers to 1. We further tuned the learning rate over $\{0.00001, 0.0001, 0.001, 0.01, 0.1\}$ and the sample size (number of nodes selected per batch) over $\{1500, 1750, 2000, 2250\}$, except for the large graphs, for which we set the sample size to $5,000$.

We maintained consistency across all methods by employing identical parameters for maximum epochs and early stopping criteria. Specifically, we set the maximum number of epochs to $30,000$ and utilized an early stopping patience of $20$ epochs, with the exception of large datasets, where we extended the patience to $500$ epochs.

In each case, we selected the hyperparameters that resulted in the lowest unsupervised training loss.

### 8.3.2 Supervised Training

For all unsupervised methods, including the baselines and our method, we perform post-training supervised evaluation using logistic regression with 60 Optuna trials. We set the maximum number of epochs to 10000 and select the epoch and hyperparameters that yield the best validation accuracy. The learning rate is swept over the range $\{0.00001, 0.0001, 0.001, 0.0015, 0.01, 0.015, 0.1, 0.5, 1, 2\}$, and the weight decay is varied over $\{10^{-5}, 10^{-4}, 10^{-3}, 10^{-2}, 10^{-1}, 0, 0.5, 1, 3\}$.

---

[2] https://github.com/PetarV-/DGI.git
[3] https://github.com/kavehhassani/mvgrl.git
[4] https://github.com/CRIPAC-DIG/GRACE.git
[5] https://github.com/YujieMo/SUGRL.git

FiGURe: Along with the hyperparameters described above, following the approach described in [17], we also tune the combination coefficients ($\alpha_i$'s) with a separate learning rate. This separate learning rate is swept over the range $\{0.00001, 0.0001, 0.001, 0.0015, 0.01, 0.015, 0.1, 0.5, 1, 2\}$. In addition, we have a coefficient for masking the incoming embeddings from each filter, which is varied between $0$ and $1$. Furthermore, these coefficients are passed through an activation layer, and we have two options: 'none' and 'exp'. When 'none' is selected, the coefficients are used directly, while 'exp' indicates that they are passed through an exponential function before being used.

FiGURe with RFF: For the experiments involving Random Fourier Features (RFF), we use the same hyperparameter ranges as mentioned above. However, we also tune the gamma parameter which is specific to RFF projections. The gamma parameter is tuned within the range $\{0.1, 0.2, 0.3, 0.4, 0.5, 0.6, 0.7, 0.8, 0.9, 1.0, 1.1, 1.2\}$.

### 8.3.3 Negative Sampling for the Identity Filter

In our implementation of $\mathbf{F}_{\text{GPRGNN}}$ or $\mathbf{F}_{\text{BernNet}}$, we follow a specific procedure for handling the filters during training and evaluation. For all filters except the identity filter ($\mathbf{I}$), we employ the negative sampling approach described in Section 6. However, the identity filter is treated differently. During training, we exclude the identity filter and only include it during evaluation.

During negative sampling, the generation of the negative anchor involves shuffling the node features, followed by premultiplying the shuffled node feature matrix with the filter matrix and computing the mean. On the other hand, for the positive anchor, the same procedure is applied without shuffling the node features. This approach encourages the model to learn meaningful patterns and relationships in the data when the filter matrix is not the identity matrix.

The decision to exclude the identity filter during training is based on the observation that it presents a special case where the positive and negative anchors become the same. As a result, the model would optimize and minimize the same quantity, potentially leading to trivial solutions. To prevent this, we exclude the identity filter during training.

By excluding the identity filter during training, we ensure that the model focuses on the other filters in $\mathbf{F}_{\text{GPRGNN}}$ or $\mathbf{F}_{\text{BernNet}}$ to capture and leverage the diverse information present in the graph. Including the identity filter only during evaluation allows us to evaluate its contribution to the final performance of the model. This approach helps prevent the model from learning trivial solutions and ensures that it learns meaningful representations by leveraging the other filters.

## 8.4 Comparison with other Supervised Methods

Table 10 presents a comparison with common supervised baselines. Specifically, we choose 3 models for comparison, representing three different kinds of supervised methods, standard aggregation models (GCN), spectral filter-based models (GPRGNN) and smart-aggregation models ($H_2$GCN). There are two key observations from this table. Firstly, FiGURe$_{512}$ is competitive with the supervised baselines, lagging behind only by a few percentage points in some cases. This suggests that much of the information that is required by the downstream tasks, captured by the supervised models, can be made available through unsupervised methods like FiGURe which uses filter banks. It is important to note that in FiGURe we only utilize logistic regression while evaluating on the downstream task. This is much more efficient that training a graph neural network end to end. Additionally it is possible that further gains may be obtained by utilizing a non-linear model like an MLP. Furthermore, as indicated by 10, we can gain further computational efficiency by utilizing lower dimensional representations like 32 and 128 (with RFF), and still not compromise significantly on the performance. Overall FiGURe manages to remain competitive despite not having access to task-specific labels and is computationally efficient as well.

## 8.5 Increasing the depth of the Encoder

We present an analysis in Table 11 featuring an increased number of encoder layers. In this context, "encoder layers" can be interpreted in two ways: firstly, as a deeper GCN which entails aggregating information from multiple-hop neighborhoods into the node; and secondly, as a single-hop GCN with a more extensive network for feature transformation. We provide performance results for both scenar-

Table 10: Contains node classification accuracy percentages on heterophilic and homophilic datasets. GCN, GPRGNN and H$_2$GCN are supervised methods. FiGURe$_{32}^{\text{RFF}}$ and FiGURe$_{128}^{\text{RFF}}$ refer to FiGURe trained with 32 and 128 dimensional representations, respectively, and then projected using RFF. The remaining models are trained at 512 dimensions. Higher numbers indicate better performance.

| | HETEROPHILIC DATASETS | | | | | HOMOPHILIC DATASETS | | | |
| | SQUIRREL | CHAMELEON | ROMAN-EMPIRE | MINESWEEPER | ARXIV-YEAR | OGBN-ARXIV | CORA | CITESEER | PUBMED |
|---|---|---|---|---|---|---|---|---|---|
| GCN | 47.78 (2.13) | 62.83 (1.52) | 73.69 (0.74) | 89.75 (0.52) | 46.02 (0.26) | 69.37 (0.00) | 87.36 (0.91) | 76.47 (1.34) | 88.41 (0.46) |
| GPRGNN | 46.31 (2.46) | 62.59 (2.04) | 64.85 (0.27) | 86.24 (0.61) | 45.07 (0.21) | 68.44 (0.00) | 87.77 (1.31) | 76.84 (1.69) | 89.08 (0.39) |
| H$_2$GCN | 37.90 (2.02) | 58.40 (2.77) | 60.11 (0.52) | 89.71 (0.31) | 49.09 (0.10) | OOM | 87.81 (1.35) | 77.07 (1.64) | 89.59 (0.33) |
| FiGURe$_{32}^{\text{RFF}}$ | 48.89 (1.55) | 65.66 (2.52) | 67.67 (0.77) | 85.28 (0.71) | 41.30 (0.21) | 66.58 (0.00) | 82.56 (0.87) | 71.25 (2.20) | 84.18 (0.53) |
| FiGURe$_{128}^{\text{RFF}}$ | 48.78 (2.48) | 66.03 (2.19) | 68.10 (1.09) | 85.16 (0.58) | 41.94 (0.15) | 69.11 (0.00) | 86.14 (1.13) | 73.34 (1.91) | 85.41 (0.52) |
| FiGURe | 52.23 (1.19) | 68.55 (1.87) | 70.99 (0.52) | 85.58 (0.49) | 42.26 (0.20) | 69.69 (0.00) | 87.00 (1.24) | 74.77 (2.00) | 88.60 (0.44) |

ios, encompassing two and three layers each. It is noteworthy that the single-layer GCN achieves equal or superior performance compared to all other configurations.

Table 11: Analyzing the impact of altering the number of encoder layers on downstream accuracy, it is evident that in all instances, the single-layer GCN outperforms the others.

| | CORA | CITESEER | SQUIRREL | CHAMELEON |
|---|---|---|---|---|
| 1 Layer GCN | 87.00 (1.24) | 74.77 (2.00) | 52.23 (1.19) | 68.55 (1.87) |
| 2 Layer GCN | 86.62 (1.43) | 73.62 (1.46) | 43.80 (1.57) | 53.53 (2.13) |
| 3 Layer GCN | 84.40 (1.84) | 72.52 (2.09) | 42.79 (1.12) | 61.73 (2.25) |
| GCN+ 2 Layer MLP | 85.73 (1.03) | 70.21 (2.30) | 49.91 (2.68) | 68.18 (1.76) |
| GCN+ 3 Layer MLP | 84.99 (1.43) | 71.39 (2.32) | 45.85 (3.33) | 64.19 (1.43) |

## 8.6 RFF Projections

As shown in Section 6.2 and in Section 6.4, RFF projections are a computationally efficient way to achieve training by preserving the latent class behavior present in lower dimensional embeddings, by projecting them into a higher dimensional linearly separable space. The natural question that comes up is how do we compute these RFF projections? We provide an algorithm to compute the RFF projections in this section, in algorithm 1. Note that this follows [40].

---
**Algorithm 1** Random Fourier Feature Computation

---
**Require:** Input data $X \in \mathbb{R}^{N \times d}$, target dimension $D$, kernel bandwidth $\gamma$
**Ensure:** Random Fourier Features $Z \in \mathbb{R}^{N \times D}$
1: Initialize random weight matrix $W \in \mathbb{R}^{d \times D}$ with Gaussian distribution
2: Initialize random bias vector $b \in \mathbb{R}^D$ uniformly from $[0, 2\pi]$
3: Compute scaled input $X' = \gamma X W + b$
4: Compute random Fourier features $Z = \sqrt{\frac{2}{D}} \cos(X')$
5: **return** $Z$

---

## 8.7 Visualising RFF Behavior and Community Structure

As shown in prior sections, FiGURe improves on both computational efficiency as well as performance by utilising RFF projections. In this section, we aim to gain insights into the behavior of RFF projections and comprehend their underlying operations through a series of simple visualizations.

**t-SNE Plots:** Figure 3 offers insights into the structure of the embeddings for the CORA dataset across different dimensions. Remarkably, even at lower dimensions (e.g., 32 dimensions), clear class structures are discernible, indicating that the embeddings capture meaningful information related to the class labels. Furthermore, when employing RFF to project the embeddings into higher dimensions, these distinct class structures are still preserved. This suggests that the role of RFF is not to introduce new information, but rather to enhance the suitability of lower-dimensional embeddings

for linear classifiers while maintaining the underlying class-related information. Notably, even at 512 dimensions, the class structures remain distinguishable. However, it is worth noting that the class-specific embeddings appear to be more tightly clustered and less dispersed compared to the 32-dimensional embeddings or the projected 32-dimensional embeddings. This suggests that learning a 512-dimensional embedding differs inherently from learning a 32-dimensional embedding and subsequently projecting it into higher dimensions.

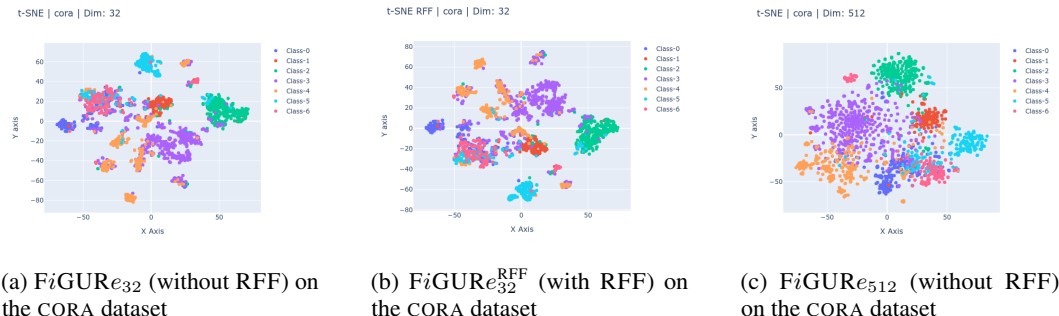

(a) F$i$GUR$e_{32}$ (without RFF) on the CORA dataset

(b) F$i$GUR$e_{32}^{\mathrm{RFF}}$ (with RFF) on the CORA dataset

(c) F$i$GUR$e_{512}$ (without RFF) on the CORA dataset

Figure 3: The figures present t-SNE plots for the CORA dataset. These plots showcase the embeddings generated by the $\mathbf{F}_3$ filter, which corresponds to $\mathbf{A}^2$ in the case of FiGURe. The t-SNE plots are generated at different embedding dimensions, providing insights into the distribution and clustering of the embeddings for each dataset.

**Correlation Plots:** Figure 4 offers insights into the correlation patterns within the embeddings generated from the SQUIRREL dataset across different dimensions. In lower dimensions, the embeddings exhibit high correlation with each other, which can be attributed to the presence of a mixture of topics or latent classes within the dataset. However, when the embeddings are projected to higher dimensions using RFF, the correlation is reduced, and a block diagonal matrix emerges. This block diagonal structure indicates the presence of distinct classes or communities within the dataset. Even at 512 dimensions, a more refined block diagonal structure can be observed compared to the correlation matrix of the 32-dimensional embeddings. Furthermore, it is noteworthy that the correlation of the projected embeddings can be regarded as a sparser version of the correlation observed in the 512-dimensional embeddings.

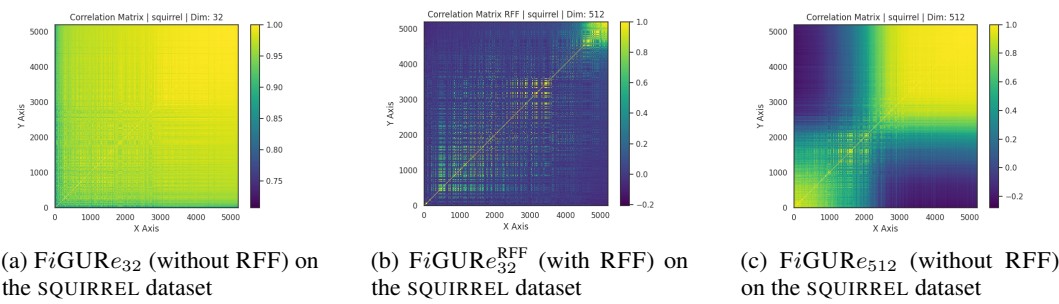

(a) F$i$GUR$e_{32}$ (without RFF) on the SQUIRREL dataset

(b) F$i$GUR$e_{32}^{\mathrm{RFF}}$ (with RFF) on the SQUIRREL dataset

(c) F$i$GUR$e_{512}$ (without RFF) on the SQUIRREL dataset

Figure 4: The figures display the normalized correlation plots for the SQUIRREL dataset. These plots illustrate the normalized correlation values between embeddings generated by the $\mathbf{F}_3$ filter. In the case of FiGURe, this filter corresponds to the square of the adjacency matrix ($\mathbf{A}^2$). The normalized correlation provides a measure of similarity or agreement between the embeddings obtained using the $\mathbf{F}_3$ filter for different embedding dimensions. These plots can help analyze the consistency or variation of embeddings across different dimensions and datasets.

## 8.8 Other Random Projections

We conducted an analysis exploring the impact of utilizing random projections beyond RFF in Table 12. Our findings reveal that RFF outperforms alternative random projection methods.

Table 12: Accuracy (%) comparison of different random projections with RFF. Notably, RFF consistently outperforms the other random projection methods.

| Features | CORA | CITESEER | SQUIRREL | CHAMELEON |
|---|---|---|---|---|
| Polynomial (d=2) [21] | 81.35 (2.11) | 69.81 (2.14) | 38.57 (1.56) | 53.88 (1.94) |
| Polynomial (d=10) [21] | 80.44 (1.56) | 68.71 (1.67) | 38.08 (1.10) | 55.20 (1.97) |
| Exp [21] | 80.42 (2.14) | 68.81 (1.34) | 38.06 (1.61) | 55.50 (2.11) |
| ANOVA [3] | 83.26 (0.78) | 70.09 (2.44) | 40.77 (1.46) | 56.01 (1.86) |
| RFF | 87.00 (1.24) | 74.77 (2.00) | 52.23 (1.19) | 68.55 (1.87) |

## 8.9 RFF ablation study at $128$ dimensions

We replicate the analysis conducted in 6.2 for 128 dimensions, as outlined in Table 13. It is apparent that our approach consistently outperforms or matches most other methods when operating at 32 dimensions across various datasets. This trend persists even after projecting the learned embeddings to higher dimensions through RFF. Our experiment reaffirms the observations detailed in 6.2.

Table 13: Node classification accuracy percentages with and without using Random Fourier Feature projections (on 128 dimensions). A higher number means better performance. The performance is improved by using RFF in almost all cases, indicating the usefulness of this transformation.

| | RFF | CORA | CITESEER | SQUIRREL | CHAMELEON |
|---|---|---|---|---|---|
| DGI | × | 84.99 (1.36) | 72.22 (2.50) | 34.22 (1.47) | 49.82 (2.96) |
| | ✓ | 84.17 (2.11) | 72.65 (1.52) | 37.97 (1.41) | 57.72 (2.03) |
| MVGRL | × | 85.31 (1.66) | 73.42 (1.63) | 36.92 (1.04) | 55.20 (1.70) |
| | ✓ | 84.61 (1.74) | 72.81 (2.13) | 38.73 (1.22) | 57.81 (1.80) |
| SUGRL | × | 71.49 (1.15) | 63.85 (2.27) | 38.04 (1.17) | 53.03 (1.73) |
| | ✓ | 71.40 (1.40) | 63.06 (2.22) | 43.24 (1.63) | 57.04 (1.78) |
| GRACE | × | 80.87 (1.49) | 62.52 (3.57) | 41.25 (1.32) | 63.14 (1.89) |
| | ✓ | 79.70 (1.91) | 64.47 (2.12) | 52.29 (1.81) | 68.90 (2.05) |
| FiGURe$_{128}$ | × | 84.73 (1.13) | 73.07 (1.13) | 41.06 (1.51) | 59.08 (3.36) |
| | ✓ | 86.14 (1.13) | 73.34 (1.91) | 48.78 (2.48) | 66.03 (2.19) |

## 8.10 RFF ablation study at $512$ dimensions

We present the results obtained by applying RFF to 512 dimensions, as shown in Table 14. It is evident that there is minimal performance improvement when using RFF on embeddings that are already of sufficient dimensionality.

Table 14: FiGURe at 512 dimensions

| | CORA | CITESEER | SQUIRREL | CHAMELEON |
|---|---|---|---|---|
| RFF@512 | 86.84 (0.98) | 74.40 (1.30) | 51.86 (1.87) | 68.60 (1.57) |

## 8.11 Issues with including RFF in training

In the context of incorporating Random Feature techniques like RFF into machine learning models, some challenges have arisen in the training process. In the Transformer attention module, these features can exhibit large variances in zero kernel score regions, leading to unstable training [8]. Additionally, increasing the sampling rate in methods like Performer and Random Feature Attention (RFA) [35] can also introduce instability, as highlighted in [39]. Another area where the use of random features with deep networks has been explored is kernel learning. There are of course, computational and convergence challenges that have been observed with these methods as well [11][12]. FiGURe avoids these problems by excluding the random feature generation from training altogether. Parameters like gamma, involved in the random feature construction, are treated as hyperparameters and a search is performed over them. Kernel learning, however, is an important area of research, and integrating RFF in the training loop would be an interesting extension of this work.

## 8.12 Computational Comparisons with other Other Unsupervised Methods

Table 15: Mean epoch time (in milliseconds) averaged across 20 trials with different hyperparameters. A lower number means the method is faster. Even though our method is slower at 512 dimensions, using 128 and 32 dimensional embeddings significantly reduces the mean epoch time. Using RFF as described in 6.2 we are able to prevent the performance drops experienced by SUGRL and GRACE.

|  | SUGRL | GRACE | FiGURe$_{512}$ | FiGURe$_{128}^{\text{RFF}}$ | FiGURe$_{32}^{rff}$ |
|---|---|---|---|---|---|
| CORA | 15.92 (4.10) | 51.19 (6.8) | 114.38 (0.51) | 20.10 (0.46) | 11.54 (0.34) |
| CITESEER | 24.37 (4.92) | 77.16 (7.2) | 156.24 (0.56) | 30.30 (0.60) | 17.16 (0.51) |
| SQUIRREL | 33.63 (6.94) | 355.2 (67.34) | 257.65 (0.76) | 47.72 (1.40) | 23.52 (1.14) |
| CHAMELEON | 16.91 (5.90) | 85.05 (14.1) | 98.36 (0.64) | 18.56 (0.39) | 11.63 (0.48) |

In Section 6.4, we compared the computational time of FiGURe with MVGRL and DGI, as all three methods fall under the category of unsupervised methods that preform contrastive learning with representations of the entire graph. However, there is another class of methods, such as SUGRL and GRACE, that contrast against other nodes without the need for graph representation computation. Consequently, these methods exhibit higher computational efficiency. Hence, as show in Table 15 upon initial inspection, it appears that SUGRL (at 512 dimensions) exhibits the highest computational efficiency, even outperforming $FiGURe_{128}^{\text{RFF}}$. However, despite its computational efficiency, the significant drop in performance across datasets (as discussed in Section 6.1) renders it less favorable for consideration. In fact, $FiGURe_{32}^{\text{RFF}}$ offers computational cost savings compared to SUGRL, while also achieving significantly better downstream classification accuracy. Turning to GRACE, it demonstrates greater computational efficiency than FiGURe$_{512}$ for low to medium-sized graphs. However, as the graph size increases, due to random node feature level masking and edge level masking, the computational requirements of GRACE substantially increase (as evidenced by the results on SQUIRREL). Therefore, for larger graphs with more than approximately 5,000 nodes, FiGURe proves to be more computationally efficient than GRACE (even at 512 dimensions). Furthermore, considering the performance improvements exhibited by FiGURe, it is evident that FiGURe (combined with RFF projections) emerges as the preferred method for unsupervised contrastive learning in graph data.

## 8.13 Time and storage cost

In Table 16, we present a comparison of training time and storage cost between our method and DGI. To ensure fairness in the evaluation, we maintained consistent settings across all baselines. The maximum number of epochs was uniformly set to 30,000, with an early stopping patience of 20 epochs, except for larger datasets where we extended the patience to 500 epochs. It's important to note that due to varying hyperparameter tuning among the baselines, normalizing the number of epochs posed some challenges. Nevertheless, we ensured that a sufficiently high number of epochs were employed to facilitate convergence. Our approach builds upon DGI and exhibits linear scaling of training time with the number of filters used. Currently, our model employs three filters for training, resulting in a training time three times longer than that of DGI. Specifically for OGBN-ARXIV, we provide information on the number of epochs, mean epoch time, and total training time. Additionally, we report the storage cost of representations generated by these methods. Please keep in mind that the linear relationship in training time between DGI and FiGURe is not apparent here, as it factors in considerations such as batching and sampling time.

Table 16: We provide the number of epochs, mean epoch time, and total training time for OGBN-ARXIV. We also report the storage cost of the representations from these methods.

| Model / Dims | Num Epochs | Mean Epoch Time | Total time | Storage |
|---|---|---|---|---|
| DGI / 512 | 3945 | 0.77s | 50.75mins | 330.75MB |
| FiGURe / 512 | 4801 | 0.92s | 73.62mins | 1.32GB |
| FiGURe / 128 | 4180 | 0.74s | 51.55mins | 330.75MB |
| FiGURe / 32 | 3863 | 0.72s | 46.36mins | 82.69MB |

## 8.14 Choice of Filter Banks

In Section 4.1, we explore the flexibility of FiGURe to accommodate various filter banks. When making a choice, it is crucial to examine the intrinsic properties of the filters contained within different filter banks. We pick two filter banks $\mathbf{F}_{\text{BERNNET}}$ and $\mathbf{F}_{\text{GPRGNN}}$ and provide an overview of the filters contained in the filter banks. We use these two filter banks as examples to illustrate what should one be looking for, while choosing a filter bank.

**Bernstein Polynomials**: Figure 5 illustrates that as the number of Bernstein Basis increases, the focus on different parts of the eigenspectrum also undergoes changes. With an increase in polynomial order, two notable effects can be observed. Firstly, the number of filters increases, enabling each filter to focus on more fine-grained eigenvalues. This expanded set of polynomial filters allows for a more detailed examination of the eigenspectrum. Secondly, if we examine the first and last Bernstein polynomials, we observe an outward shift in their shape. This shift results in the enhancement of a specific fine-grained part at the ends of the spectrum. These observations demonstrate that Bernstein polynomials offer the capability to selectively target and enhance specific regions of interest within the eigenspectrum

**Standard Basis**: Figure 5 reveals two key observations. Firstly, at a polynomial order of 2, the standard basis exhibit focus at the ends of the spectrum, in contrast to the behavior of Bernstein polynomials, which tend to concentrate more on the middle of the eigenspectrum. This discrepancy highlights the distinct characteristics and emphasis of different polynomial bases in capturing different parts of the eigenspectrum. Secondly, as the number of polynomials increases (in contrast to Bernstein polynomials), the lower order polynomials remain relatively unchanged. Instead, additional polynomials are introduced, offering a more fine-grained focus at the ends of the spectrum. This expansion of polynomials allows for a more detailed exploration of specific regions of interest within the the ends of eigenspectrum.

In the context of filter banks, previous studies [28, 7] have demonstrated that certain datasets, such as SQUIRREL and CHAMELEON, benefit from frequency response functions that enhance the tail ends of the eigenspectrum. This observation suggests that the standard basis, which naturally focuses on the ends of the spectrum, may outperform Bernstein basis functions at lower orders. However, as the order of the Bernstein basis increases, as discussed in 4.1, there is a notable improvement in performance. This can be attributed to the increased focus of Bernstein basis functions on specific regions, particularly the ends of the spectrum. As a result, higher-order Bernstein filters exhibit enhanced capability in capturing important information in those regions. It is worth noting that the choice between $\mathbf{F}_{\text{GPRGNN}}$ and $\mathbf{F}_{\text{BERNNET}}$ depends on the specific requirements of the downstream task. If the task necessitates a stronger focus on the middle of the spectrum or requires a band-pass or comb-like frequency response, $\mathbf{F}_{\text{BERNNET}}$ is likely to outperform $\mathbf{F}_{\text{GPRGNN}}$. Thus, the selection of the appropriate filter bank should be based on the desired emphasis on different parts of the eigenspectrum. Regarding the performance comparison between $\mathbf{F}_{\text{BERNNET}}$ and $\mathbf{F}_{\text{GPRGNN}}$, it is plausible that as we increase the order of the Bernstein basis, the performance could potentially match that of $\mathbf{F}_{\text{GPRGNN}}$. However, further investigation and experimentation are required to determine the specific conditions and orders at which this convergence in performance occurs.

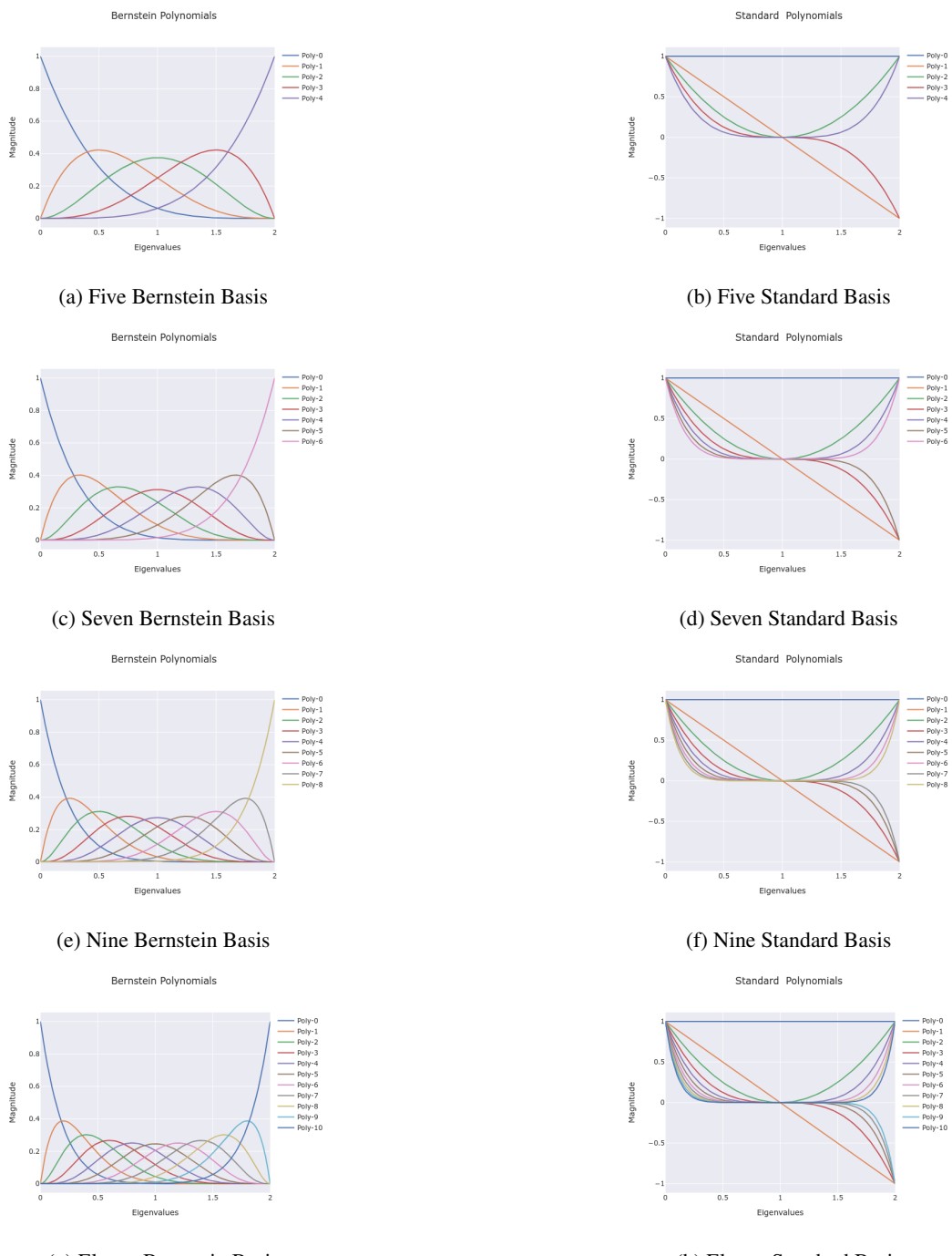

(a) Five Bernstein Basis

(b) Five Standard Basis

(c) Seven Bernstein Basis

(d) Seven Standard Basis

(e) Nine Bernstein Basis

(f) Nine Standard Basis

(g) Eleven Bernstein Basis

(h) Eleven Standard Basis

Figure 5: The figures contain the Bernstein basis as well as standard basis for different degrees. The x-axis of the figures represents the eigenvalues of the Laplacian matrix, while the y-axis represents the magnitude of the polynomials. It is important to note that while plotting the standard polynomials, they are computed with respect to the Laplacian matrix ($\mathbf{L_n}$) rather than the adjacency matrix. As a result, the eigenvalues lie between $[0, 2]$. On the other hand, the Bernstein polynomials are typically defined for the normalised Laplacian matrix, and therefore there is no change in the eigenvalue range (the eigenvalues of the normalised Laplacian matrix typically range from 0 to 2). By using the Laplacian matrix as the basis for plotting the polynomials, we can observe the behavior and magnitude of the polynomials at different eigenvalues, providing insights into their spectral properties and frequency response characteristics.

