exist, capable of projecting lower-dimensional representations into higher dimensions. These projections enhance the adaptability of these representations for downstream tasks. Random Fourier features (RFF) [31] provide a prime example of such techniques**.

## 5  Proposed Approach

The following section delineates the process of unsupervised representation learning. Post that, we give details on how the representations learned from each filter bank is used in downstream tasks using random feature maps.

## 5.1 Unsupervised Representation Learning

Our method FiGURe (**Fi**lter-based **G**raph **U**nsupervised **Re**presentation Learning) builds on concepts introduced in [11, 35], extending the maximization of mutual information between node and global filter representations for each filter in the filter bank $\mathbf{F} = \{\mathbf{F}_1, \mathbf{F}_2, ...\mathbf{F}_K\}$. We construct an encoder for each filter to maximize the mutual information between the input data and encoder output. For the $i^{\text{th}}$ filter, we learn an encoder, $E_\theta : \mathcal{X}_i \to \mathcal{X}'_i$, denoted by learnable parameters $\theta$. In this context, $\mathcal{X}_i$ represents a set of examples, where each example $[\widehat{\mathbf{X}_{ij}}, \widehat{\mathbf{F}_{ij}}] \in \mathcal{X}_i$ consists of a filter $\mathbf{F}_i$, its corresponding nodes and node features drawn from an empirical probability distribution $\mathbb{P}_i$, which captures the joint distribution of features and node representations $[\mathbf{X}, \mathbf{F}_i]$. $\mathcal{X}_i$ defines the set of representations learnt by the encoder on utilizing feature information as well as topological information from the samples, sampled from the joint distribution $\mathbb{P}_i$. The goal, aligned with [21, 11, 35], is to identify $\theta$ that maximizes mutual information between $[\mathbf{X}, \mathbf{F}_i]$ and $E_\theta(\mathbf{X}, \mathbf{F}_i)$, or $\mathcal{I}_i([\mathbf{X}, \mathbf{F}_i], E_\theta(\mathbf{X}, \mathbf{F}_i))$. While exact mutual information (MI) computation is unfeasible due to unavailable exact data and learned representations distributions, we can estimate the MI using the Jensen-Shannon MI estimator [5, 25], defined as:

$$
\begin{aligned}
\mathcal{I}^{\text{JSD}}_{i,\theta,\omega}([\mathbf{X}, \mathbf{F}_i], E_\theta(\mathbf{X}, \mathbf{F}_i)) := & \; \mathbb{E}_{\mathbb{P}_i}[-\text{sp}(T_{\theta,\omega}([\widehat{\mathbf{X}_{ij}}, \widehat{\mathbf{F}_{ij}}], E_\theta(\widehat{\mathbf{X}_{ij}}, \widehat{\mathbf{F}_{ij}}))] - \\
& \mathbb{E}_{\mathbb{P}_i \times \tilde{\mathbb{P}}_i}[\text{sp}(T_{\theta,\omega}([\widetilde{\mathbf{X}_{ij}}, \widetilde{\mathbf{F}_{ij}}], E_\theta(\widehat{\mathbf{X}_{ij}}, \widehat{\mathbf{F}_{ij}}))]
\end{aligned}
\tag{1}
$$

Here, $T_\omega : \mathcal{X}_i \times \mathcal{X}'_i \to \mathbb{R}$ represents a discriminator function with learnable parameters $\omega$. Note that $[\tilde{\mathbf{X}}_{ij}, \tilde{\mathbf{F}}_{ij}]$ is an input sampled from $\tilde{\mathbb{P}}_i$, which is a marginal of the joint distribution of the input data and the learned node representations. The function $\text{sp}(.)$ corresponds to the softplus function [6]. Additionally, $T_{\theta,\omega} = D_w \circ (\mathcal{R}(E_\theta(\widehat{\mathbf{X}_{ij}}, \widehat{\mathbf{F}_{ij}})), E_\theta(\widehat{\mathbf{X}_{ij}}, \widehat{\mathbf{F}_{ij}}))$, where $\mathcal{R}$ denotes the readout function responsible for summarizing all node representations by aggregating and distilling information into a global filter representation.

In our approach, we first obtain node representations by feeding the filter-specific topology and associated node features into the encoder: $\mathbf{H}_i = E_\theta(\mathbf{X}_i, \mathbf{F}_i) = \{h_1^{\mathbf{F}_i}, h_2^{\mathbf{F}_i}, ..., h_n^{\mathbf{F}_i}\}$. To obtain global representations, we employ a readout function $\mathcal{R} : \mathbb{R}^{N \times d'} \to \mathbb{R}^{d'}$, which combines and distills information into a global representation $h_g^{F_i} = \mathcal{R}(\mathbf{H}_i) = \mathcal{R}(E_\theta(\mathbf{X}, \mathbf{F}_i))$. Instead of directly maximizing the mutual information between the local and global representations, we introduce a learnable discriminator $D_\omega : \mathbb{R}^{d'} \times \mathbb{R}^{d'} \to \mathbb{R}$, where $D_\omega(.,.)$ represents the joint probability score between the global representation and the node-specific patch representation. This joint probability score should be higher when considering global and local representations obtained from the same filter, as opposed to the joint probability score between the global representation from one filter and the local representation from an arbitrary filter.

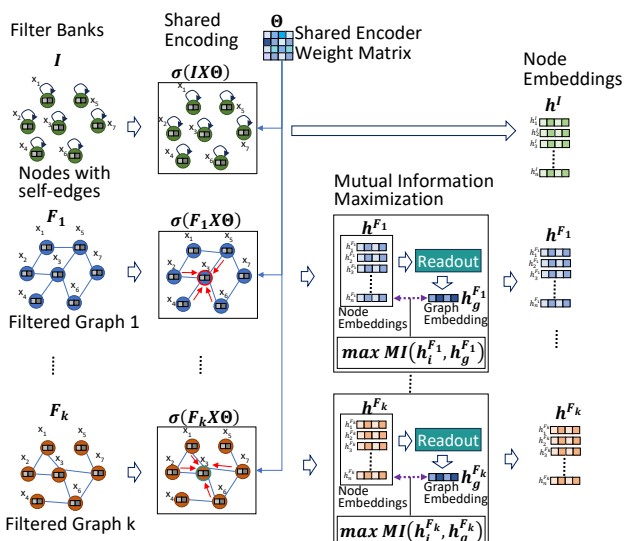

Figure 1: Unsupervised learning of node embeddings by maximizing mutual information between node and graph representations over the graphs from the filter bank. Note that the parameter $\Theta$ is shared across all the filters.

To generate negative samples for contrastive learning, we employ a corruption function $\mathcal{C} : \mathbb{R}^{N \times d} \times \mathbb{R}^{N \times N} \to \mathbb{R}^{M \times d} \times \mathbb{R}^{M \times M}$, which yields corrupted samples denoted as $[\widetilde{\mathbf{X}_{ij}}, \widetilde{\mathbf{F}_{ij}}] = \mathcal{C}(\mathbf{X}, \mathbf{F}_i)$. The designed corruption function generates data decorrelated with the input data.

In order to learn representations across all filters in the filter bank, we aim to maximise the average estimate of mutual information (MI) across all filters, considering $K$ filters.

$$\mathcal{I}_{\mathbf{F}} = \frac{1}{K} \sum_{i=1}^{K} \mathcal{I}_{i,\theta,\omega}^{JSD}([\mathbf{X}, \mathbf{F}_i], E_\theta(\mathbf{X}, \mathbf{F}_i)) \tag{2}$$

Maximising the Jenson-Shannon MI estimator is equivalent to reducing the binary cross entropy loss defined between positive samples (sampled from the joint) and the negative samples (sampled from the product of marginals). Therefore, for each filter, we minimise the following objective:

$$\mathcal{L}_{\mathbf{F}_i} = \frac{1}{N+M} \left( \sum_{j=1}^{N} \mathbb{E}_{(\mathbf{X}, \mathbf{F}_i)}[\log(D_\omega(h_j^{\mathbf{F}_i}, h_g^{\mathbf{F}_i}))] + \sum_{j=1}^{M} \mathbb{E}_{(\tilde{\mathbf{X}}, \bar{\mathbf{F}_i})}[\log(D_\omega(\tilde{h}_j^{\mathbf{F}_i}, h_g^{\mathbf{F}_i}))] \right) \tag{3}$$

Therefore to learn meaningful representations across all filters the following objective is minimised:

$$\mathcal{L} = \frac{1}{K} \sum_{i=1}^{K} \mathcal{L}_{\mathbf{F}_i} \tag{4}$$

However, managing the computational cost of training and storage for large graphs with separate node representations for each filter presents a significant challenge, exacerbated by the high dimensional requirements of contrastive learning methods. We implement parameter sharing to mitigate the first issue, borrowing the concept from studies such as [4, 10], thereby sharing the encoder's parameters $\theta$ and the discriminator's parameters $\omega$ across all filters. Instead of storing dense filter-specific node representations, we only store the parameters of the shared encoder and the first-hop neighbourhood information of each node per filter, which has a lower storage cost. For downstream tasks, we retrieve the embeddings by reconstructing filter-specific representations. To ensure quick and efficient reconstruction, we use a simple one-layer GNN. This on-demand reconstruction of filter-specific representations significantly reduces the computational and storage requirements associated with individual node representations. Fig 1 illustrates such a simple encoder's mutual information-based learning process.

Addressing the second issue, we initially train our models to generate low-dimensional embeddings. These encapsulate latent classes, as discussed in [2] as a superset of classes pertinent to downstream tasks. Although the low-dimensional embeddings harbour latent class information, they lack linear separability. Hence, we project these embeddings into a higher-dimensional space using random Fourier feature (RFF) projections, a strategy inspired by kernel methods (Section 4.2). Using this approach allows for improved linear separability of the latent classes. Our experimental findings (Section 6.2) affirm the effectiveness of projecting lower-dimensional embeddings into higher dimensions, confirming the retention of latent class information in these embeddings.

## 5.2 Supervised Representation Learning

After obtaining representations for each filter post the reconstruction of the node representations, learning an aggregation mechanism to combine information from representations that capture different parts of the eigenspectrum for the given task is necessary. We adopt learning schemes proposed in [4, 10, 20], where we learn a weighted combination of filter-specific representations. Therefore, the combined representations we learn for the downstream task are as follows (considering $K$ filters from the filter bank $\mathbf{F}$):

$$Z = \sum_{i=1}^{K} \alpha_i \phi'(E_\theta(\mathbf{X}, \mathbf{F}_i)) \tag{5}$$

The parameters $\alpha_i$'s are learnable. Additionally, the function $\phi(.)'$ represents either the RFF projection or an identity transformation, depending on whether $E_\theta(\mathbf{X}, \mathbf{F}_i)$ is low-dimensional or not. A classifier model (e.g. logistic regression) consumes these embeddings, where we train both the $\alpha_i$'s and the weights of the classifier. Fig 2 illustrates this process. The main distinction between semi-supervised methods such as [20, 4, 10] and our method is that the semi-supervised methods learn both the encoder and the combination coefficients based on labelled data. However, we pre-train the encoder in our method and subsequently learn a task-specific combination of filter-specific representations.

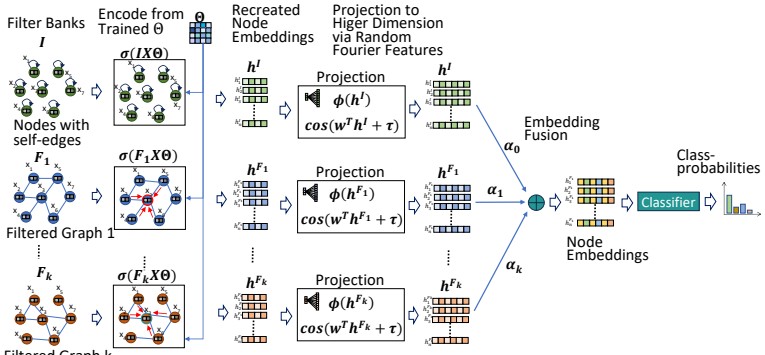

Figure 2: Supervised Learning: Using the trained parameter $\Theta$, we generate the node embeddings by encoding the filtered graphs that get consumed in the classification task.

## 6 Experimental Results

**Training Details**: We define a single-layer graph convolutional network (GCN) with shared weights ($\Theta$) across all filters in the filter bank ($\mathbf{F}$) as our encoder. Therefore, the encoder can be expressed as follows: $E_\theta(\mathbf{X}, \mathbf{F}_i) = \sigma(\mathbf{F}_i\mathbf{X}\Theta)$. It is important to note that $\mathbf{F}_i$ represents a normalized filter with self-loops, which ensures that its eigenvalues are within the range of [0, 2]. The non-linearity function $\sigma$ refers to the parametric rectified linear unit (PReLU) [9]. As we work with a single graph, we obtain the positive samples by sampling nodes from the graph. Using these sampled nodes, we construct a new adjacency list that only includes the edges between these sampled nodes in filter $\mathbf{F}_i$. On the other hand, the corruption function $\mathcal{C}$ operates on the same sampled nodes. However, it randomly shuffles the node features instead of perturbing the adjacency list. Similar to [35], we employ a straightforward readout function that involves averaging the representations across all nodes for a specific filter $\mathbf{F}_i$: $\mathcal{R}(\mathbf{H}_i) = \sigma\left(\frac{1}{N}\sum_{j=0}^{N} h_j^{\mathbf{F}_i}\right)$ where $\sigma$ denotes the sigmoid non-linearity. We utilize a bilinear scoring function, whose parameters are also shared across all filters:

$$D_\omega(h_j^{\mathbf{F}_i}, h_g^{\mathbf{F}_i}) = \sigma(h_j^{\mathbf{F}_i T}\mathbf{W}h_g^{\mathbf{F}_i}) \tag{6}$$

We learn the encoder and discriminator parameters by optimising Eq. 4. While we could use various filter banks, we specifically employ the filter bank corresponding to GPRGNN ($\mathbf{F}_{\text{GPRGNN}}$) for all our experiments. However, we also conduct an ablation study (see 6.5) to compare the performance when using $\mathbf{F}_{\text{GPRGNN}}$ versus $\mathbf{F}_{\text{BERNNET}}$. For more detailed training information, please refer to the supplementary material.

We conducted a series of comprehensive experiments to evaluate the effectiveness and competitiveness of our proposed model compared to SOTA models and methods. These experiments address the following research questions: **[RQ1]** How does FiGURe, perform compared to SOTA unsupervised models? **[RQ2]** Can we perform satisfactorily even with lower dimensional representations using projections such as RFF? **[RQ3]** Does shared encoder decrease performance? **[RQ4]** What is the computational efficiency gained by using lower dimensional representations compared to methods that rely on higher dimensional representations? **[RQ5]** Can alternative filter banks be employed to recover good quality representations?

**Datasets and Setup:** We evaluated our model on a diverse set of real-world datasets, which include both heterophilic and homophilic networks, to assess its effectiveness. Similar to previous works, we utilized the node classification task as a proxy to evaluate the quality of the learned representations. Please refer to the supplementary material for detailed information about the benchmark datasets.

The heterophilic datasets used in our evaluation include **CHAMELEON**, **SQUIRREL**, **ROMAN-EMPIRE**, and **MINESWEEPER**. For CHAMELEON and SQUIRREL, we adopted the ten random splits (with 48%, 32%, and 20% of nodes allocated for the train, validation, and test sets, respectively) from [27]. For ROMAN-EMPIRE and MINESWEEPER, we used the ten random splits provided in [30]. Additionally, we evaluated our model on four homophilic datasets: **CORA**, **CITESEER**, and **PUBMED**, as borrowed from [14]. We report the mean and standard deviation of the test accuracy across different splits. Please refer to the supplementary material for detailed statistics of each dataset.

**Baselines:** In our comparison against baselines, we considered common unsupervised approaches, such as DEEPWALK and NODE2VEC, and state-of-the-art mutual information-based methods, namely DGI, MVGRL, GRACE, and SUGRL. We also include the performance numbers of the widely used GCNfor reference. It is important to note that unless explicitly mentioned, we set the representation size to 512 dimensions for all reported results, consistent with previous work. Please refer to the supplementary material for detailed comparisons with other supervised methods and the link to our codebase.

## 6.1 RQ1: FiGURe versus SOTA Methods

Table 1: Contains node classification accuracy percentages on homophilic and heterophilic datasets. FiGURe$_{32}$ and FiGURe$_{128}$ refer to FiGURe trained with 32 and 128 dimensional representations, respectively, and then projected using RFF. The remaining models are trained at 512 dimensions. Higher numbers indicate better performance. It is worth noting that FiGURe achieves superior performance or remains competitive with the baseline methods in all cases. The rightmost column Av. $\Delta_{gain}$ represents the average accuracy % gain of FiGURe over the model in that row, averaged across the different datasets. Blue, Red and Green represent the 1$^{st}$, 2$^{nd}$ and 3$^{rd}$ best performing models, for a particular dataset.

| | HETEROPHILIC DATASETS | | | | HOMOPHILIC DATASETS | | | |
|---|---|---|---|---|---|---|---|---|
| | SQUIRREL | CHAMELEON | ROMAN-EMPIRE | MINESWEEPER | CORA | CITESEER | PUBMED | Av. $\Delta_{gain}$ |
| DEEPWALK | 38.66 (1.44) | 53.42 (1.73) | 13.08 (0.59) | 79.96 (0.08) | 83.64 (1.85) | 63.66 (3.36) | 80.85 (0.44) | 16.35 |
| NODE2VEC | 42.60 (1.15) | 54.23 (2.30) | 12.12 (0.30) | 80.00 (0.00) | 78.19 (1.14) | 57.45 (6.44) | 73.24 (0.59) | 18.56 |
| DGI | 39.61 (1.81) | 59.28 (1.23) | 47.54 (0.76) | 82.51 (0.47) | 84.57 (1.22) | 73.96 (1.61) | 86.57 (0.52) | 7.67 |
| MVGRL | 39.90 (1.39) | 54.61 (2.29) | 68.50 (0.38) | 85.60 (0.35) | 86.22 (1.30) | 75.02 (1.72) | 87.12 (0.35) | 4.39 |
| GRACE | 53.15 (1.10) | 68.25 (1.77) | 47.83 (0.53) | 80.22 (0.45) | 84.79 (1.51) | 67.60 (2.01) | 87.04 (0.43) | 5.54 |
| SUGRL | 43.13 (1.36) | 58.60 (2.04) | 39.40 (0.49) | 82.40 (0.58) | 81.21 (2.07) | 67.50 (1.62) | 86.90 (0.54) | 9.80 |
| FiGURe$_{32}$ | 48.89 (1.55) | 65.66 (2.52) | 67.67 (0.77) | 85.28 (0.71) | 82.56 (0.87) | 71.25 (2.20) | 84.18 (0.53) | 3.18 |
| FiGURe$_{128}$ | 48.78 (2.48) | 66.03 (2.19) | 68.10 (1.09) | 85.16 (0.58) | 86.14 (1.13) | 73.34 (1.91) | 85.41 (0.52) | 2.11 |
| FiGURe | 52.23 (1.19) | 68.55 (1.87) | 70.99 (0.52) | 85.58 (0.49) | 87.00 (1.24) | 74.77 (2.00) | 88.60 (0.44) | 0.00 |

Table 2: Comparison of Node classification accuracy percentages with the widely used supervised model GCN. Despite not having access to task specific labels, FiGURe learns good quality representations.

| | SQUIRREL | CHAMELEON | ROMAN-EMPIRE | MINESWEEPER | CORA | CITESEER | PUBMED |
|---|---|---|---|---|---|---|---|
| GCN | 47.78 (2.13) | 61.43 (2.70) | **73.69 (0.74)** | **89.75 (0.52)** | **87.36 (0.91)** | **76.47 (1.34)** | 88.41 (0.46) |
| FiGURe | **52.23 (1.19)** | **68.55 (1.87)** | 70.99 (0.52) | 85.58 (0.49) | 87.00 (1.24) | 74.77 (2.00) | **88.60 (0.44)** |

We analyzed the results in Table 1 and made important observations. Across homophilic and heterophilic datasets, FiGURe consistently outperforms several SOTA unsupervised models, except in a few cases where it achieves comparable performance. We want to emphasize the rightmost column of the table, which shows the average percentage gain in performance across all datasets. This metric compares the improvement that FiGURe provides over each baseline model for each dataset and averages these improvements. This metric highlights the performance consistency of FiGURe across diverse datasets. No other baseline model achieves the same consistent performance across all datasets as FiGURe. Even the recent state-of-the-art contrastive models GRACE and SUGRL experience average performance drops of approximately 5% and 10%, respectively. This result indicates that FiGURe learns representations that exhibit high generalization and task-agnostic capabilities. Another important observation is the effectiveness of RFF projections in improving lower dimensional representations. We compared FiGURe at different dimensions, including F$i$GUR$e_{32}$ and F$i$GUR$e_{128}$, corresponding to learning 32 and 128-dimensional embeddings, respectively, in addition to the baseline representation size of 512 dimensions. Remarkably, even at lower dimensions, FiGURe with RFF projections demonstrates competitive performance across datasets, surpassing the 512-dimensional baselines in several cases. This result highlights the effectiveness of RFF projections in enhancing the quality of lower dimensional representations. Section 6.2 discusses more insights about the effectiveness of RFF projections. Furthermore, we include the widely used supervised model, GCN, in Table 2 as a benchmark for comparison. Notably, FiGURe outperforms GCN on heterophilic datasets, except for ROMAN-EMPIRE and MINESWEEPER, while achieving competitive performance on homophilic datasets. Please refer to supplementary material for detailed comparisons with supervised methods.

## 6.2 RQ2: RFF Projections on Lower Dimensional Representations

Table 3: Node classification accuracy percentages with and without using Random Fourier Feature projections (on 32 dimensions). A higher number means better performance. The performance is improved by using RFF in almost all cases, indicating the usefulness of this transformation

|  | RFF | CORA | CITESEER | SQUIRREL | CHAMELEON |
|---|---|---|---|---|---|
| DGI | × | **81.65 (1.90)** | 65.62 (2.39) | 31.60 (2.19) | 45.48 (3.02) |
|  | ✓ | 81.49 (1.96) | **66.50 (2.44)** | **38.19 (1.52)** | **56.01 (2.66)** |
| MVGRL | × | 78.81 (1.73) | 70.36 (1.76) | 29.58 (0.94) | 46.56 (2.84) |
|  | ✓ | **80.14** (2.41) | **70.57** (1.56) | **37.83** (1.32) | **55.57 (2.28)** |
| SUGRL | × | 65.35 (2.41) | 42.84 (2.57) | 31.62 (1.47) | 43.20 (1.79) |
|  | ✓ | **70.06** (1.24) | **47.03** (3.02) | **38.50 (2.19)** | **51.01 (2.26)** |
| GRACE | × | 76.84 (1.09) | 58.40 (3.05) | 38.20 (1.38) | 53.25 (1.58) |
|  | ✓ | **79.15 (1.44)** | **63.66 (2.96)** | **51.56 (1.39)** | **67.39 (2.23)** |
| FiGURe | × | **82.88 (1.42)** | 70.32 (1.98) | 39.38 (1.35) | 53.27 (2.40) |
|  | ✓ | 82.56 (0.87) | **71.25 (2.20)** | **48.89 (1.55)** | **65.66 (2.52)** |

In this section, we analyse the performance of unsupervised baselines using 32-dimensional embeddings with and without RFF projections (see Table 3). Despite extensive hyperparameter tuning, we could not replicate the results reported by SUGRL, so we present the best results we obtained. Two noteworthy observations emerge from these tables. Firstly, it is evident that lower dimensional embeddings can yield meaningful and linearly separable representations when combined with simple RFF projections. Utilising RFF projections enhances performance in almost all cases, highlighting the value captured by MI-based methods even with lower-dimensional embeddings. Secondly, FiGURe consistently achieves superior or comparable performance to the baselines, even in lower dimensions. Notably, this includes SUGRL, purported to excel in such settings. However, there is a 2-3% performance gap between GRACE and our method for the SQUIRREL and CHAMELEON datasets. While GRACE handles heterophily well at lower dimensions, its performance deteriorates with homophilic graphs, unlike FiGURe which captures lower frequency information effectively. Additionally, our method exhibits computational efficiency advantages for specific datasets in lower dimensions. Please refer to the supplementary material for more details. Overall, these findings highlight the potential of RFF projections in extracting useful information from lower dimensional embeddings and reaffirm the competitiveness of FiGURe over the baselines.

## 6.3 RQ3: Sharing Weights Across Filter Specific Encoders

Table 4: A comparison of the performance on the downstream node classification task using independently trained encoders and weight sharing across encoders is shown. The reported metric is accuracy. In both cases, the embeddings are combined using the method described in 5.2

|  | CORA | CITESEER | SQUIRREL | CHAMELEON |
|---|---|---|---|---|
| INDEPENDENT | 86.92 (1.10) % | 75.03 (1.75) % | 50.52 (1.51) % | 66.86 (1.85) % |
| SHARED | 87.00 (1.24) % | 74.77 (2.00) % | 52.23 (1.19) % | 68.55 (1.87) % |

Our method proposes to reduce the computational load by sharing the encoder weights across all filters. It stands to reason whether sharing these weights causes any degradation in performance. We present the results with shared and independent encoders across the filters in Table 4 to verify this. The findings indicate no significant decrease in performance when using shared weights, and in some cases, it even leads to improvements, validating the use of shared encoders.

## 6.4 RQ4: Computational Efficiency

To assess the computational efficiency of the different methods, we analyzed the computation time and summarized the results in Table 5. The key metric used in this analysis is the mean epoch time: the average time taken to complete one epoch of training. We compared our method with other MI based methods such as DGI and MVGRL. Due to the increase in the number of augmentation views,

Table 5: Mean epoch time (in milliseconds) averaged across 20 trials with different hyperparameters. A lower number means the method is faster. Even though our method is slower at 512 dimensions, using 128 and 32 dimensional embeddings significantly reduces the mean epoch time. Using RFF as described in 6.2 we are able to prevent the performance drops experienced by DGI and MVGRL.

|  | DGI | MVGRL | FiGURe | FiGURe$_{128}$ | FiGURe$_{32}$ |
|---|---|---|---|---|---|
| CORA | 38.53 (0.77) | 75.29 (0.56) | 114.38 (0.51) | 20.10 (0.46) | 11.54 (0.34) |
| CITESEER | 52.98 (1.15) | 102.41 (0.99) | 156.24 (0.56) | 30.30 (0.60) | 17.16 (0.51) |
| SQUIRREL | 87.06 (2.07) | 168.24 (2.08) | 257.65 (0.76) | 47.72 (1.40) | 23.52 (1.14) |
| CHAMELEON | 33.08 (0.49) | 64.71 (1.05) | 98.36 (0.64) | 18.56 (0.39) | 11.63 (0.48) |

there is an expected increase in computation time from DGI to MVGRL to FiGURe. However, as demonstrated in 6.2, using RFF projections allows us to achieve competitive performance even at lower dimensions. Therefore, we also included comparisons with our method at 128 and 32 dimensions in the table. It is evident from the results that our method, both at 128 and 32 dimensions, exhibits faster computation times compared to both DGI and MVGRL, which rely on higher-dimensional representations to achieve good performance. This result indicates that FiGURe is computationally efficient due to its ability to work with lower-dimensional representations. During training, our method, F$i$GURe$_{32}$, is $\sim$ 3x faster than DGI and $\sim$ 6x times faster than MVGRL. Despite the faster computation, F$i$GURe$_{32}$ also exhibits an average performance improvement of around 2% across the datasets over all methods considered in our experiments. Please refer to the supplementary material for additional comparisons to other unsupervised models.

## 6.5 RQ5: Experiments on Other Filter Banks

Table 6: Accuracy percentage results using other filter banks for FiGURe. $\mathbf{F}^3_{\text{BERNNET}}$ refers to the $\mathbf{F}_{\text{BERNNET}}$ filter bank (Section 4.1) with $K$ set to 3 and $\mathbf{F}^{11}_{\text{BERNNET}}$ refers to $K$ set to 11.

|  | CORA | CITESEER | SQUIRREL | CHAMELEON |
|---|---|---|---|---|
| $\mathbf{F}^3_{\text{BERNNET}}$ | 85.13 (1.26) | 73.38 (1.81) | 37.07 (1.29) | 53.95 (2.78) |
| $\mathbf{F}^{11}_{\text{BERNNET}}$ | 86.62 (1.59) | 73.97 (1.43) | 43.48 (3.80) | 62.13 (3.66) |
| $\mathbf{F}_{\text{GPRGNN}}$ | 87.00 (1.24) | 74.77 (2.00) | 52.23 (1.19) | 68.55 (1.87) |

To showcase the versatility of our proposed framework, we conducted an experiment using Bernstein filters, as detailed in Table 6. The results indicate that using $\mathbf{F}_{\text{GPRGNN}}$ leads to better performance than Bernstein filters. We believe that the reason this is happening is due to the latent characteristics of the dataset. [10, 20] have shown that datasets like CHAMELEON and SQUIRREL need frequency response functions that give more prominence to the tail-end spectrum. $\mathbf{F}_{\text{GPRGNN}}$ are more amenable to these needs, as demonstrated in [20]. However, datasets requiring frequency response similar to comb filters may be better approximated by $\mathbf{F}_{\text{BERNNET}}$ as their basis gives uniform prominence on the entire spectrum. Please refer to the supplementary material, which shows the basis frequency responses of these two filter banks, with more clarification. Therefore, although $\mathbf{F}_{\text{GPRGNN}}$ gives better performance for these datasets, there could be datasets where $\mathbf{F}_{\text{BERNNET}}$ could do better. Hence, we proposed a general framework that can work with any filter bank.

## 7 Conclusion and Future Work

Our work demonstrates the benefits of enhancing contrastive learning methods with filter views and learning filter-specific representations to cater to diverse tasks from homophily to heterophily. We have effectively alleviated computational and storage burdens by sharing the encoder across these filters and focusing on low-dimensional embeddings that utilize high-dimensional projections, a technique inspired by random feature maps developed for kernel approximations. Future directions include extending the analysis in [2] to graph contrastive learning and explicitly exploring the linear separability in low dimensions. This analysis could solidify the connection with the proposed random feature maps approach.

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

### 8.4 Evaluation on large graphs

Table 8: Contains node classification accuracy percentages on two large-scale datasets OGBN-ARXIV and ARXIV-YEAR have been added. FiGURe$_{32}$ and FiGURe$_{128}$ refer to FiGURe trained with 32 and 128 dimensional representations, respectively, and then projected using RFF. The remaining models are trained at 512 dimensions. Higher numbers indicate better performance. Blue, Red and Green represent the $1^{\text{st}}$, $2^{\text{nd}}$ and $3^{\text{rd}}$ best performing models, for a particular dataset.

| | HETEROPHILIC DATASETS | | | | | HOMOPHILIC DATASETS | | | | |
| | SQUIRREL | CHAMELEON | ROMAN-EMPIRE | MINESWEEPER | ARXIV-YEAR | CORA | CITESEER | PUBMED | OGBN-ARXIV | Av. $\Delta_{gain}$ |
|---|---|---|---|---|---|---|---|---|---|---|
| DGI | 39.61 (1.81) | 59.28 (1.23) | 47.54 (0.76) | 82.51 (0.47) | 40.59 (0.09) | 84.57 (1.22) | 73.96 (1.61) | 86.57 (0.52) | 65.58 (0.00) | 6.61 |
| MVGRL | 39.90 (1.39) | 54.61 (2.29) | 68.50 (0.38) | 85.60 (0.35) | OOM | 86.22 (1.30) | 75.02 (1.72) | 87.12 (0.35) | OOM | 4.39 |
| GRACE | 53.15 (1.10) | 68.25 (1.77) | 47.83 (0.53) | 80.22 (0.45) | OOM | 84.79 (1.51) | 67.60 (2.01) | 87.04 (0.43) | OOM | 5.54 |
| SUGRL | 43.13 (1.36) | 58.60 (2.04) | 39.40 (0.49) | 82.40 (0.58) | 36.96 (0.19) | 81.21 (2.07) | 67.50 (1.62) | 86.90 (0.54) | 65.80 (0.00) | 8.64 |
| FiGURe$_{32}$ | 48.89 (1.55) | 65.66 (2.52) | 67.67 (0.77) | 85.28 (0.71) | 41.30 (0.21) | 82.56 (0.87) | 71.25 (2.20) | 84.18 (0.53) | 66.58 (0.00) | 3.18 |
| FiGURe$_{128}$ | 48.78 (2.48) | 66.03 (2.19) | 68.10 (1.09) | 85.16 (0.58) | 41.94 (0.15) | 86.14 (1.13) | 73.34 (1.91) | 85.41 (0.52) | 69.11 (0.00) | 2.11 |
| FiGURe | 52.23 (1.19) | 68.55 (1.87) | 70.99 (0.52) | 85.58 (0.49) | 42.26 (0.20) | 87.00 (1.24) | 74.77 (2.00) | 88.60 (0.44) | 69.69 (0.00) | 0.00 |

Similar to Table 1, Table 8, provides a comparison with SOTA methods such as DGI, MVGRL, GRACE and SUGRL. However, in this table we also incorporate the large-scale datasets ARXIV-YEAR and OGBN-ARXIV. FiGURe shows good performance on these datasets as well, demonstrating the scalability of our method. The last column, average percentage gain, is updated accordingly. Two baseline methods MVGRL and GRACE run into memory issues on the larger datasets and are accordingly reported OOM in the table. Even the lower-dimensional representations (with RFF projections) are able to beat the baselines on these large scale datasets. Overall, FiGURe is consistently able to provide gains over the baselines methods regardless of the kind of graph, homophilic, heterophilic or large-scale. This once again demonstrates the generalizability of FiGURe. It is noteworthy that computational efficiency gained by reducing the dimension size becomes significant with the scale of the dataset. On ARXIV-YEAR for example, 128 dimensional embeddings give 1.6x speedup and 32 dimensional embeddings give 1.7x speedup.