# OpenReview forum: "FiGURe: Simple and Efficient Unsupervised Node Representations with Filter Augmentations"
_NeurIPS.cc/2023/Conference — NeurIPS 2023 poster_

### Official Review · Reviewer_xu4A · 2023-06-08

**Soundness:** 3 good
**Presentation:** 3 good
**Contribution:** 3 good
**Rating:** 5
**Confidence:** 3

**Summary:**

The proposer proposed a contrastive learning method for graphs. The goal is to learn representations for nodes in an unsupervised way, by maximizing mutual informations between the local representation and global representation of a graph, training a single encoder to learn node representations. The method proposes to use multiple different filters to learn representations with different emphasis on the graph structure (as seen through the spectral representation), and then combine those representations on downstream tasks, by learning the mixing coefficients in addition to a linear classifier.

The proposed method, FIGURe, being slower than the other methods it is compared to, when learning 512 representations, the authors also explore ways to produce more compact representations, 32 and 128 in width, using RFF to recover some of the lost accuracy on downstream tasks.

**Strengths:**

* The results show improvement compared to other self-supervised approaches. Although FIGURe does not get the first place on every single benchmark, it performs very consistency across datasets, leading to the highest average performance.
* The versions of FIGURe with smaller representations are leading to good results as well, underlying the soundness of the approach followed by the authors to reduce the dimension of representation and then recovering part of the lost accuracy.
* Overall, the paper is well written, and easy to follow. The preliminary section is concise and yet very useful.

**Weaknesses:**

* The study of the authors is limited to 1-layer GCN. It would be interesting to know how the method compares to others as the GNN architecture gets bigger (in particular deeper).
* The authors justify the need for learning representations of smaller sizes due to the prohibitive cost of contrastive learning, but this part lacks justification in my view: the mean epoch time does indeed increase compared to baselines (+50%), but we are missing details in the paper related to how many epochs it is trained on (in absolute and compared to others methods) - this would help justify the need for the lower dimensionality version and the RFF tricks used afterwards
* The authors present the shared encoder as an efficiency approach, but it seems from Table 4 that it is also increasing performance.
* In general, I would appreciate having more details on the experimental section, and in particular on the training recipe and the evaluation recipes - some of these questions will be asked in the "Questions" sections of this review - as this would help understanding if the comparison are fair (and help reproducing results)

**Questions:**

Here are a bunch of details that I think need clarification in the experimental sections:
* How long is the method trained compared to the baselines ? Does the number of filter influence the training time (linearly?).
* How many filters does the method used compare to the baselines ? If it uses more, should we normalize those to make sure comparisons are fair ?
* How are the baselines evaluated ? Are those also using the learnable alpha mixing coefficients ? Are those baselines using the same number of filters ? If not, should we normalize those to make sure the evaluations are fair ?
* What happens if we use RFF on the 512 dimensions representations ? In particular, on the baselines ?
* What happens if we do not use RFF on the 32 dimensions and 128 dimensions representations ?

In several sections, the authors talk about the cost of having high number of dimensions and the cost of having several encoder instead of one shared encoder, would it be possible to put numbers on these costs in terms of time / storage ? It would help convey the need for the dimensionality reduction and RFF tricks.

---

> ### Author Rebuttal · Authors · 2023-08-10
>
> First and foremost, we would like to express our gratitude for the time and effort you dedicated to reviewing our paper, providing us with constructive feedback. We're glad to learn that you found our paper well-written, with a concise and useful preliminary section. We highly appreciate your positive remarks about FIGURe's consistent performance across datasets and its demonstrated soundness in our approach, especially with reduced dimension representations. The acknowledgment of FIGURe's improvement over other self-supervised approaches further underscores the value of our contribution.
>
> 1. **Multi Layer Encoder:** Refer to global response.
>
> 2. **Time and Storage Cost**: For all the baselines, we used the same number of maximum epochs and the same early stopping criteria. The maximum number of epochs was set to 30,000 and the early stop patience was set to 20 epochs, except for the large datasets, for which the patience was set to 500 epochs. Due to the fact that each baseline was tuned over multiple hyperparameters, each having a different convergence criterion, it would be difficult to normalize the epochs. However, we made sure that we set the epochs to a large value such that the models generally converged to a solution. We build upon DGI. The training time of our model scales linearly with the number of filters. Since our model currently uses three filters for training, our training time is three times larger than that of DGI. We provide the number of epochs, mean epoch time, and total training time for ogbn-arxiv. We also report the storage cost of the representations from these methods. Please note that that the linear time relationship between DGI and FiGURe is not visible here due to the inclusion of batching and sampling time.
>
> | Model / Dims| Num Epochs | Mean Epoch Time | Total time | Storage |
> |-----------|--------------------------|-------------------------------|--------------------------|--------------------------|
> | DGI / 512       | 3945   | 0.77s                         |      50.75mins           | 330.75 MB
> | FiGURe / 512       | 4801                     | 0.92s                         | 73.62mins                | 1.32GB|
> | FiGURe / 128       | 4180                     | 0.74s                         | 51.55mins                | 330.75 MB|
> | FiGURe / 32        | 3863                     | 0.72s                         | 46.36mins               | 82.69MB|
>
> We see both the training time and storage saved by utilizing lower dimensions along with RFF.
>
> 3. **Shared Encoder Improvements:** We hypothesize that weight sharing among encoders results in embedding different filter representations in a shared subspace, thereby enhancing their suitability for learning a combined representation. This ultimately leads to improved features for downstream tasks and, in some cases, results in performance improvements. We will add this point in the final version of the paper.
>
> 4. **Normalize Comparisons with Baselines:**
>      1. Normalized number of parameters: One way to normalize the models is to ensure that they have roughly equal number of parameters. In the FiGURe model with shared encoder, the number of parameters used are exactly the same as the number of parameters used in DGI and GRACE.  MVGRL and SUGRL use two encoder models, hence roughly have twice the number of parameters. Hence, the FiGURe model uses equal or fewer parameters.
>      2. Normalized filter inputs: Another way to normalize the inputs is to provide the same filter augmentations to all the baselines. However, there are non-trivial challenges in incorporating filter banks in the other baselines. Hence, we chose to incorporate them in DGI. We do believe that it should be possible to incorporate them in other baseline models as well and it is left for future research.
>
> 5. **RFF on 512 dimensions**: We report the result of using RFF on 512 dimensions. We notice that there is not much improvement in performance, when you already have sufficiently large dimensionsal embeddings.
> | Cora         | Citeseer     | Squirrel     | Chameleon    |
> |--------------|--------------|--------------|--------------|
>   86.84 (0.98) | 74.40 (1.30) | 51.86 (1.87) | 68.60 (1.57) |
>
> 6. **With and without RFF**:
> In Table -3, we have shown how our method performs with 32 dimensions with and without RFF. One can observe that our method is better than or comparable to most other methods at 32 dimensions across datasets. This trend continues even after projecting the learnt embeddings via RFF to larger dimensions. Additionally, we report the same table but with 128 dimensions below. Similar observations can be made here as well.
>
> |         | RFF | cora        | citeseer    | squirrel    | chameleon   |
> |---------|------------|--------------|--------------|--------------|--------------|
> | DGI     | $\times$   | 84.99 (1.36) | 72.22 (2.50) | 34.22 (1.47) | 49.82 (2.96) |
> |        | $\checkmark$ | 84.17 (2.11) | 72.65 (1.52) | 37.97 (1.41) | 57.72 (2.03) |
> | Hassani | $\times$   | 85.31 (1.66) | 73.42 (1.63) | 36.92 (1.04) | 55.20 (1.70) |
> |        | $\checkmark$ | 84.61 (1.74) | 72.81 (2.13) | 38.73 (1.22) | 57.81 (1.80) |
> | SUGRL   | $\times$   | 71.49 (1.15) | 63.85 (2.27) | 38.04 (1.17) | 53.03 (1.73) |
> |        | $\checkmark$ | 71.40 (1.40) | 63.06 (2.22) | 43.24 (1.63) | 57.04 (1.78) |
> | GRACE   | $\times$   | 80.87 (1.49) | 62.52 (3.57) | 41.25 (1.32) | 63.14 (1.89) |
> |        | $\checkmark$ | 79.70 (1.91) | 64.47 (2.12) | 52.29 (1.81) | 68.90 (2.05) |
> | FiGURe| $\times$   | 84.73 (1.13) | 73.07 (1.13) | 41.06 (1.51) | 59.08 (3.36) |
> |        | $\checkmark$ | 86.14 (1.13) | 73.34 (1.91) | 48.78 (2.48) | 66.03 (2.19) |

---

### Official Review · Reviewer_wrwF · 2023-07-06

**Soundness:** 2 fair
**Presentation:** 1 poor
**Contribution:** 2 fair
**Rating:** 3
**Confidence:** 3

**Summary:**

This work proposes a model and contrastive learrning method for acquiring a comprehensive specturm of graph representation by employing filters of various levels. They are adaptively aggregated with learnable weights for downstream supervision tasks. As a result, this approach performs well on both homophilic and heterophilic graphs. Additionally, by utilizing a random featrue map for kernel approximation, it demonstrates effectiveness with low computational cost.  They employ a shared encoder for multiple filters, which not only reduces computational load compared to using independent filters for each, but also improves performane in some cases. Experimental results exhibit performance improvement on average.

**Strengths:**

They conducted experiments on data with multiple characteristics, demonstrating that their approach works for both homophilic and heterophilic graphs. They proposed various methods for computational efficiency and provide evidence of the effectiveness through experiments. The contribution of their work is clearly described through abtract and introduction sections.

**Weaknesses:**

This paper requires some improvements in terms of presentation, as it contains inconsistent notation (such as size of tilde and hat in the line 147 and 149) and deferred explanation for newly introduced symbols that is difficult to follow and numerous mathematical errors. The presence of these mathematical errors prompted a desire to check the implementation, which unfortunately was not provided by the authors. If these issues are addressed in the rebuttal and the necessary improvements are made, it would be considered for increasing score. It is necessary to clarify the notation and provide better explanations to ensure that readers can follow the content more easily.

For example, in Section 5.1, the notation $D_w$ is introduced for the first time on line 149, while the explanation for it appears much later, specifically on line 166.

Additionally, there seem to be errors in the equations. In Equation 1, the correct formulation of the MI estimator should be:

$E_{P_i}[-sp(-T)] - E_{P_i \times \tilde{P}_i}[sp(T)]$.  (details are omitted here. Just refer the ‘-‘ sign)
(7th slide in [this lecture](http://people.ee.duke.edu/~lcarin/Jiachang3.20.2020.pdf))

This formulation intuitively maximizes MI, as the first term increases the similarity between local and global features obtained from the same graph, while the second term decreases the similarity between local and global features obtained from different graphs. Similar errors can be observed in Equation 3, where minimizing the given objective $\mathcal{L}_{F_i}$ seems to result in learning that decreases the similarity between local and global representations from the same graph, particularly in the first term. Moreover, in the second term of Equation 3, only the corrupted one $(\tilde{X}_i, \tilde{F}_i)$ is considered for expectation and $h_g^{F_i}$ is not covered, which, leads to ambiguous representation. Furthermore, in Section 6, $\sigma$ on lines 227, 233, and in Equation (6) refer to different functions, causing confusion.

Overall, it is crucial to address these issues related to notation, mathematical accuracy, and clarity throughout the paper to enhance its quality and ensure a better understanding for readers.

**Questions:**

In Section 6.5, the authors conducted ablation studies for filters, and it was observed that GPRGNN performed better in all cases. However, it would be informative to explore if there are any other options that demonstrate superior performance. This could involve examining alternative filter configurations or considering different methods altogether. Providing such insights would enhance the comprehensiveness of the analysis and strengthen the paper's contribution. Furthermore, analyzing the learned weight $\alpha$ or  TSNE plots of learned representations with different filters would also be interesting.

**Limitations:**

-

---

> ### Author Rebuttal · Authors · 2023-08-09
>
> We thank you for dedicating time to thoroughly review our paper and for providing comprehensive feedback on our work. We appreciate your recognition of the strengths of our research. It is encouraging to note that you found value in our experiments across data with multiple characteristics, demonstrating the versatility of our approach in handling both homophilic and heterophilic graphs. We are also glad that you acknowledged the clarity of our contributions in the abstract and introduction sections, and our methods proposed for computational efficiency. We genuinely acknowledge the concerns you have raised regarding the presentation, inconsistencies in notation, and mathematical errors.
>
> 1. **Writing improvements:** We apologize for the notational inconsistencies. We thank the reviewer for bringing this to light. We have made the requested changes so that the paper is consistent in its notations. Specifically:
>      1. The size of the tilde has been changed to be consistent with the hats in line 147 and 149.
>      2. Notation $D_{w}$ is now correctly introduced before its use in line 149.
>      3. The MI estimator in equation 1 has now been corrected $\mathcal{I}^{\textrm{JSD}}\_{i, \theta, \omega}(\cdot, \cdot):= \mathbb{E}\_{\mathbb{P}\_{i}}[-\text{sp}(-T\_{\theta, \omega}(\cdot, \cdot)]- \\ \mathbb{E}\_{\mathbb{P}\_{i} \times \mathbb{\widetilde{P}}\_{i}}[\text{sp}(T\_{\theta, \omega}(\cdot,  \cdot)]$
>      4. Equation 3 has been fixed: $\mathcal{L}\_{F\_i} = -\frac{1}{N + M}\left( \sum_{j=1}^{N}\mathbb{E}[\text{log}(D_{\omega}(h_{j}^{F\_{i}}, h_{g}^{F\_{i}}))] + \sum\_{j=1}^{M}\mathbb{E}[\text{log}(1 - D\_{\omega}(\widetilde{h\_{j}^{F\_{i}}}, h\_{g}^{F_{i}}))] \right)$
>      5. We have mitigated the confusion about the sigma variable, by using psi to represent PReLU and sigma to represent sigmoid
>      6. Additionally, we rewrote some parts to reduce the confusion
>
> 2. **Code Access:** We would like to point out that the anonymized link to the code is present in the supplementary material in line number 471, in section 8.1.
>
> 3. **Additional Analysis**: We appreciate your suggestion to broaden our exploration and consider other options as well. In addition to the BernNet filters reported in Section 6.5 of the paper, we report the performance of the Chebyshev filters used in ChebNet [A]. We report two configurations, 3 filters and 11 filters, similar to the analysis done with BernNet filters. As can be observed from the table, the GPRGNN filters outperform this too. It has been shown in previous works [B] that heterophilic datasets like Chameleon and Squirrel require filters that focus on the tail ends of the spectrum, leading to a dumbell/parabolic frequency response function. In the case of the GPRGNN filters the model is able to learn this response function easily, by giving a high weightage to the $A^{2}$ filter. Similarly, homophilic datasets require low-pass filters, which the GPRGNN filters can easily produce by giving higher weightage to the A and $A^3$ filters. However the coefficients required to create these shape using other filter banks such as BernNet and ChebNet, although possible, are not so simple and the model has a hard time learning them. We believe that this is the reason why GPRGNN filters give better performance on the datasets that we have considered. However, we would like to point out that there may be datasets where other frequency response shapes are required, and in these cases it is possible that BernNet and ChebNet filters are more suited to the problem. \\
> Additionally, we report the alpha coefficients of the best performing split using GPRGNN filters, on Cora, Citeseer, Squirrel and Chameleon (The coefficients will be slightly different for different splits in the same dataset). The observations from this table support our hypothesis about the coefficients of the GPRGNN filters. Specifically, it can be observed that the homophilic datasests Cora and Citeseer are $A^{3}$ dominant, leading the creation of a low pass filters. In contrast, for the heterophilic datasets Chameleon and Squirrel, the $A^{2}$ filter is dominant, creating a parabolic filter response function. We will add TSNE plots for different filters and filter-banks to the final version of the paper.
> * [A]: Convolutional Neural Networks on Graphs with Chebyshev Approximation, Revisited. NeurIPS 2022.
> * [B]:  A Piece-wise Polynomial Filtering Approach for Graph Neural Networks, ECML PKDD 2022

---

> > ### Author Response · Authors · 2023-08-10
> > **Additional Analysis (Tables)**
> >
> > Please find below, the Tables from the "Additional Analysis" section of the rebuttal:
> >
> > **Alternative Filter Banks Analysis**
> >
> > |                          | Cora        | Citeseer    | Squirrel    | Chameleon    |
> > |--------------------------|--------------|--------------|--------------|---------------|
> > | $\mathbf{F}_{BernNet}^3$    | 85.13 (1.26) | 73.38 (1.81) | 37.07 (1.29) | 53.95 (2.78)  |
> > | $\mathbf{F}_{BernNet}^{11}$ | 86.62 (1.59) | 73.97 (1.43) | 43.48 (3.80) | 62.13 (3.66)  |
> > | $\mathbf{F}_{ChebNet}^3$ | 83.84 (1.36) | 71.92 (2.29) | 40.23 (1.58) | 60.61 (2.03)  |
> > | $\mathbf{F}_{ChebNet}^{11}$ | 76.14 (6.80) | 59.89 (8.94) | 52.46 (1.10) | 67.37 (1.60)  |
> > | $\mathbf{F}_{GPRGNN}$       | 87.00 (1.24) | 74.77 (2.00) | 52.23 (1.19) | 68.55  (1.87) |
> >
> >
> > **Alpha Coefficient Analysis**
> >
> > |           | I    | $A$ | $A^{2}$ | $A^{3}$  |
> > |-----------|------|---|------|-------|
> > | Cora      | 18.2 | 0 | 0    | 35.95 |
> > | Citeseer  | 0    | 0 | 0    | 0.48  |
> > | Squirrel  | 0    | 0 | 15.3 | 0     |
> > | Chameleon | 0    | 0 | 8.93 | 0.1   |

---

### Official Review · Reviewer_Ck44 · 2023-07-06

**Soundness:** 3 good
**Presentation:** 2 fair
**Contribution:** 2 fair
**Rating:** 6
**Confidence:** 3

**Summary:**

The paper proposes a contrastive learning model for learning node embeddings on a graph, with two technical innovations: First, the authors propose a new augmentation scheme during contrastive learning. Secondly, the authors re-map high dimensional embeddings into lower dimensional space using random Fourier features. The authors claim improvements attributed to both technical innovations on a variety of datasets.

**Strengths:**

Presentation if the paper is mostly clear, figures are well readable, and results are clearly presented.

**Weaknesses:**

- Although the word "augmentation" is very present in both the paper title and abstract (6 occurences), it only appears once in the paper. It is not fully clear how the abstract claims map onto the technique in the paper, beyond the preliminary info in Section 4.1. The terms should be used more consistently here. It is somewhat inferable from 4.1 what the "augmentations" are, but this needs to be clarified in Section 5.
- Presentation in section 5 is a bit unclear to me. The algorithm proposes two innovations (filter augmentations and random Fourier features), but the experiment section is not grouped accordingly. I would propose to split the content in 5.1. and 5.2. differently to aid the understanding of the different algorithmic components and their implementation.
- l. 183 claims "Maximizing JS is equiv. to reducing BCE" --- can you add a suitable reference for this claim?
- **Baselines**: I checked a few of the reported baselines numbers. For instance, [SUGRL (Table 1)](https://ojs.aaai.org/index.php/AAAI/article/view/20748) reports $83.5 \pm .5$ on Cora vs. $81.21 \pm 2.07$ here, $73.0 \pm 0.4$ on CiteSeer vs. $67.5 \pm 1.62$ here. Also some other baseline numbers seem to be off. Could you comment on possible discrepancies, and whether the code you are running for your experiment can reproduce the numbers reported in the literature? Some of the differences I mentioned above are similar in scale to the ones you use to claim the effectiveness of your method.
- From Table 3 (which I find convincing), it seems like RFF substantially improves many of the baseline methods. It is not really clear to me whether the reported gains are mostly due to the proposed augmentation, or mainly due to the RFF. A clean ablation experiement, ideally on all the datasets would help. Specifically, is the method SOTA also *without* RFF, i.e., due to the different representation learning approach? This is not addressed by the experiments currently.
- In general, the experimental results seem limited. See my clarification questions below.

**Minor:**

- The choice of colors in Table 1 is probably not well readable by readers with colorblindness.
- The word "significant" is used multiple times in the text, but no statistical tests were run. I propose to either add statistics to your tables (my recommendation), or drop/replace the term.

**Questions:**

- l. 178: What is the corruption function?
- Eq. 3: I think the equation is off and misses two minus signs, once in front of the loss term, and one in front of the negative examples. I.e., shouldnt the general form be $ - \left(\sum \mathbb E \log D_\omega(.) + \sum \mathbb E \log (1 - D_\omega(.)) \right)$ ?
- What is the rationale for only running experiments on all datasets in Table 1, vs. for all the tables? Also, could you outline the choice of these particular datasets for your study? For instance, SUGRL also evaluates on Ogbn-arxiv, Ogbn-mag, Ogbn-products.
- Could you more clearly outline your contribution w.r.t. the representation learning part? The loss formulation as such seems to be established, still a substantial amount of space in section 5 is distributed to discussing the loss and its derivation. My impression is that mainly the composition and augmentation of the positive pair distinguishes this part from previous work. Could you confirm/discuss this further?
- Does the FIGURe model in Table 1 use RFF, or not? If it does not use RFF as suggested by the caption, then the result in Table 1 would contradict the result in Table 3. Could you clarify?
- Are there any assumptions underlying your proposed augmentation scheme with respect to the properties of the datasets? I.e., are the classification tasks in which a representation obtained using FIGURe can be expected to perform worse than other techniques due to the augmentation design?

---

> ### Author Rebuttal · Authors · 2023-08-09
>
> We thank you for taking the time to review our paper. We truly appreciate your insightful feedback. We are glad to know that you found the presentation of our paper mostly clear, with readable figures and clearly presented results.
>
> # Comment Response
>
> 1. **Explaining Augmentation:** We acknowledge the inconsistency in the usage of the term "augmentation" and will strive to enhance clarity by using the term consistently and providing a more detailed explanation in Section 5.
>
> 2. **Numerical Differences:** Refer to global response.
>
> 2. **Presentation/RFF Gains/Filter Gains:** In Table 2, FiGURe refers to the model with 512 dimensional embeddings without RFF, whereas FiGUR$e_{32}$ and FiGUR$e_{128}$ refer to models with lower dimensional embeddings that utilize RFF. No single baseline does consistently well across all heterophilic and homophilic datasets. Comparing FiGURe with other methods shows that using only filter augmentation is consistently either competitive or doing better than existing methods. Note that all the baseline methods are utilizing the same number of dimensions as FiGURe. FiGUR$e_{32}$ and FiGUR$e_{128}$ show the value of RFF to improve performance and be competitive even when learning lower dimensional embeddings. So the way to read the gains from the individual innovation parts is: 1] Filter augmentation provides consistent performance across all datasets which is comparable or better than all baselines and 2] RFF allows for efficient training via lower dimensions while providing similar gains as training with large dimensional embeddings. We will add more clarification and rearrange the tables to get the message across better.
>
> 3. **Max JS implies Min BCE Reference:** We realized that this statement is imprecise. Maximizing the JS MI estimator is a hard problem but it can be approximatelly optimized as a BCE loss following the works in [A,B,C]
>
> 4. **Choice of Colors:** Thank you for making us aware of this. We will fix this in the final version of the paper.
>
> 5. **Clarity of Corruption Function:** Refer to global response.
>
> 6. **Equation Discrepancy:** Refer to global response.
>
> 7. **Choice of Dataset:** SUGRL primarily focused on homophilic datasets, hence we selected a few datasets with varying levels of homophily [D], to demonstrate the capability of our method. Our ablation studies showed results on two homophilic and two heterophilic datasets, demonstrating some of the properties we want to showcase, with regards to our method. We also show results on ArXiv-Year and OGBN-ArXiv to show that we can scale up to 150k+ nodes. Please refer to Table - 8 in the supplementary material. We plan to conduct additional experiments on ogbn-mag/products which have 1M+ nodes.
>
> 8. **Contribution Summary:** One of the main contributions of this work is the notion of filter augmentations. Prior works have incorporated various filter banks in the supervised setting, and they jointly learn the filter specific representations and the combination coefficients. The novel insight that we are proposing in this paper is that we can separately learn these two parts and **bridge the gap with supervised methods**. This enabled us to propose a **simple extension to DGI** which easily incorporates these filter banks. However, augmenting more filters can increase the training cost. To save on compute, we also show that reducing the size of the learnt embeddings and then projecting them using RFF allows one to achieve similar performance with lower dimensional embeddings.
>
> 9. **Assumptions and threats to model:** Our model is based on previous filter-based supervised models that have been proposed in the literature. It has been shown in previous works [E] that heterophilic datasets like Chameleon and Squirrel require filters that focus on the tail ends of the spectrum, leading to a dumbell/parabolic frequency response function. In the case of the GPRGNN filters the model is able to learn this response function easily, by giving a high weightage to the $A^{2}$ filter. Similarly, homophilic datasets require low-pass filters, which the GPRGNN filters can easily produce by giving higher weightage to the A and/or $A^3$ filters. We show some of the learnt coefficients in the table below for the two homophilic and heterophilic datasets.
>
> |           | I    | $A$ | $A^{2}$ | $A^{3}$  |
> |-----------|------|---|------|-------|
> | Cora      | 18.2 | 0 | 0    | 35.95 |
> | Citeseer  | 0    | 0 | 0    | 0.48  |
> | Squirrel  | 0    | 0 | 15.3 | 0     |
> | Chameleon | 0    | 0 | 8.93 | 0.1   |
>
> However, the coefficients required to create these shapes using other filter banks such as BernNet and ChebNet, although possible, are not so simple and the model has a hard time learning them. We believe that this is the reason why GPRGNN filters give better performance on the datasets that we have considered. However, we would like to point out that there may be datasets where other frequency response shapes are required, and in these cases, it is possible that other filter banks like BernNet/ChebNet filters are more suited to the problem.
>
>
> Additionally, we would like to point out that DGI allowed us to easily extend and incorporate filter banks, while other methods like MVGRL, GRACE and SUGRL, provide some non-trivial challenges to incorporate them. However, we do believe that it would be possible, and would like to include this as a part of future research.
>
> * [A] Learning deep representations by mutual information estimation and maximization, ICLR 2019
> * [B] Learning Independent Features with Adversarial Nets for Non-linear ICA, ICML 2017 Workshop on Implicit Models
> * [C] f-GAN: Training Generative Neural Samplers using Variational Divergence Minimization, NeurIPS 2016
> * [D] Geom-GCN: Geometric Graph Convolutional Networks, ICLR 2020
> * [E] A Piece-wise Polynomial Filtering Approach for Graph Neural Networks, ECML PKDD 2022

---

> > ### Comment · Reviewer_Ck44 · 2023-08-17
> > **Re: Baselines**
> >
> > Thank you for the response. I'll reply to the other points in a separate comment, but wanted to get back regarding the baselines you mentioned in your global comment.
> >
> > I understand the reason for choosing the protocol from [A, B, C], but still have trouble to reference the respective numbers. If possible, I would appreciate a table where you cite numbers from the respective papers alongside your reproduced numbers for an easy overview of how the literature vs. reproduced numbers add up.
> >
> > For instance, this would be great to have for all SOTA methods in Table 1:
> >
> > | | SQUIRREL | CHAMELEON |  ROMAN-EMPIRE  | MINESWEEPER |  CORA | CITESEER  | PUBMED |
> > | - | - | -| - | -| - | -| - |
> > |DeepWalk ([...], Table X) | | | | | | |
> > |DeepWalk (reproduced) | | | | | | |
> > |... | | | | | | |
> > |... | | | | | | |
> >
> > where "reproduced" would be the numbers already in Table 1 in the submission --- this would help me to quickly trace back the sources. (I realize that in some settings there might be some mismatch). Thanks for considering.

---

> > > ### Author Response · Authors · 2023-08-17
> > >
> > > Thank you for your response.
> > >
> > > Please find the table below. As mentioned above (in the global response), we reuse the splits utilized in [A,B,C] which are different from the splits used in the original papers of the baselines we compare against [D,E,F,G]. However, as requested we are adding the numbers from [D,E,F,G], but please note they are not strictly comparable.
> > >
> > > Note that in the leftmost column, the naming convention is as follows - Method Name (Reference, Table-X) (Tab-X denotes Table-X). For example: DGI (E, Tab-2) denotes Method DGI, Reference E, Table-2. The references are given at the bottom. We would like to clarify that for all the baseline methods, we ran the code provided by the authors of those works along with extensive hyperparameter tuning. Specifically, we separate the hyperparameter tuning of the unsupervised part and supervised part, and do not use any information from the supervised part while tuning the hyperparameters of the unsupervised part.
> > >
> > >
> > > |                        | Squirrel     | Chameleon    | Roman-Empire | Minesweeper  | ArXiv-Year   | Cora         | Citeseer     | Pubmed       | OGBN-ArXiv   |
> > > |------------------------|--------------|--------------|--------------|--------------|--------------|--------------|--------------|--------------|--------------|
> > > |                        |              |              |              |              |              |              |              |              |              |
> > > | Deepwalk (D, Tab-2)    |              |              |              |              |              | 67.2         | 43.2         | 65.3         |              |
> > > | Deepwalk (E, Tab-2)    |              |              |              |              |              | 70.7 ± 0.6   | 51.4 ± 0.5   | 74.3 ± 0.9   |              |
> > > | Deepwalk (F, Tab-1)    |              |              |              |              |              | 75.7         | 50.5         | 80.5         |              |
> > > | Deepwalk (G, Tab-1,2)    |              |              |              |              |              | 67.2 ± 0.2   | 43.2 ± 0.4   | 65.3 ± 0.5   | 63.6 ± 0.4   |
> > > | Deepwalk (Reproduced, Tab-1) | 38.66 (1.44) | 53.42 (1.73) | 13.08 (0.59) | 79.96 (0.08) |              | 83.64 (1.85) | 63.66 (3.36) | 80.85 (0.44) |              |
> > > |                        |              |              |              |              |              |              |              |              |              |
> > > | Node2Vec (F, Tab-1)    |              |              |              |              |              | 74.8         | 52.3         | 80.3         |              |
> > > | Node2Vec (Reproduced, Tab-1) | 42.60 (1.15) | 54.23 (2.30) | 12.12 (0.30) | 80.00 (0.00) |              | 78.19 (1.14) | 57.45 (6.44) | 73.24 (0.59) |              |
> > > |                        |              |              |              |              |              |              |              |              |              |
> > > | DGI (D, Tab-2)         |              |              |              |              |              | 82.3 ± 0.6   | 71.8 ± 0.7   | 76.8 ± 0.6   |              |
> > > | DGI (E, Tab-2)         |              |              |              |              |              | 82.3 ± 0.6   | 71.8 ± 0.7   | 76.8 ± 0.6   |              |
> > > | DGI (F, Tab-1)         |              |              |              |              |              | 82.6 ± 0.4   | 68.8 ± 0.7   | 86.0 ± 0.1   |              |
> > > | DGI (G, Tab-1, 2)         |              |              |              |              |              | 82.3 ± 0.5   | 71.5 ± 0.4   | 79.4 ± 0.3   | 65.1 ± 0.4   |
> > > | DGI (Reproduced, Tab-1)      | 39.61 (1.81) | 59.28 (1.23) | 47.54 (0.76) | 82.51 (0.47) | 40.59 (0.09) | 84.57 (1.22) | 73.96 (1.61) | 86.57 (0.52) | 65.58 (0.00) |
> > > |                        |              |              |              |              |              |              |              |              |              |
> > > | MVGRL (E, Tab-2)       |              |              |              |              |              | 86.8 ± 0.5   | 73.3 ± 0.5   | 80.1 ± 0.7   |              |
> > > | MVGRL (G, Tab-1,2)       |              |              |              |              |              | 82.9 ± 0.3   | 72.6 ± 0.4   | 80.1 ± 0.7   | 68.7 ± 0.4   |
> > > | MVGRL (Reproduced, Tab-1)    | 39.90 (1.39) | 54.61 (2.29) | 68.50 (0.38) | 85.60 (0.35) | OOM          | 86.22 (1.30) | 75.02 (1.72) | 87.12 (0.35) | OOM          |
> > > |                        |              |              |              |              |              |              |              |              |              |
> > >
> > > Continued below.

---

> > > > ### Author Response · Authors · 2023-08-17
> > > >
> > > > Continued:
> > > >
> > > >
> > > > |                        | Squirrel     | Chameleon    | Roman-Empire | Minesweeper  | ArXiv-Year   | Cora         | Citeseer     | Pubmed       | OGBN-ArXiv   |
> > > > |------------------------|--------------|--------------|--------------|--------------|--------------|--------------|--------------|--------------|--------------|
> > > > |                        |              |              |              |              |              |              |              |              |              |
> > > > | GRACE (F, Tab-1)       |              |              |              |              |              | 83.3 ± 0.4   | 72.1 ± 0.5   | 86.7 ± 0.1   |              |
> > > > | GRACE (G, Tab-1,2)       |              |              |              |              |              | 83.1 ± 0.2   | 72.1 ± 0.1   | 79.6 ± 0.5   | 68.7 ± 0.4   |
> > > > | GRACE (Reproduced, Tab-1)    | 53.15 (1.10) | 68.25 (1.77) | 47.83 (0.53) | 80.22 (0.45) | OOM          | 84.79 (1.51) | 67.60 (2.01) | 87.04 (0.43) | OOM          |
> > > > |                        |              |              |              |              |              |              |              |              |              |
> > > > | SUGRL (G, Tab-2)       |              |              |              |              |              | 83.4 ± 0.5   | 73.0 ± 0.4   | 81.9 ± 0.3   | 88.9 ± 0.2   |
> > > > | SUGRL (Reproduced, Tab-1,2)    | 43.13 (1.36) | 58.60 (2.04) | 39.40 (0.49) | 82.40 (0.58) | 36.96 (0.19) | 81.21 (2.07) | 67.50 (1.62) | 86.90 (0.54) | 65.80 (0.00) |
> > > > |                        |              |              |              |              |              |              |              |              |              |
> > > > | FIGURE$_{32}$              | 48.89 (1.55) | 65.66 (2.52) | 67.67 (0.77) | 85.28 (0.71) | 41.30 (0.21) | 82.56 (0.87) | 71.25 (2.20) | 84.18 (0.53) | 66.58 (0.00) |
> > > > | FIGURE$_{128}$             | 48.78 (2.48) | 66.03 (2.19) | 68.10 (1.09) | 85.16 (0.58) | 41.94 (0.15) | 86.14 (1.13) | 73.34 (1.91) | 85.41 (0.52) | 69.11 (0.00) |
> > > > | FIGURE$_{512}$             | 52.23 (1.19) | 68.55 (1.87) | 70.99 (0.52) | 85.58 (0.49) | 42.26 (0.20) | 87.00 (1.24) | 74.77 (2.00) | 88.60 (0.44) | 69.69 (0.00) |
> > > > ---
> > > >
> > > > ## References
> > > >
> > > > [A] Geom-GCN: Geometric Graph Convolutional Networks, ICLR 2020
> > > >
> > > > [B] A Piece-wise Polynomial Filtering Approach for Graph Neural Networks, ECML PKDD 2022
> > > >
> > > > [C] A critical look at the evaluation of GNNs under heterophily: are we really making progress? ICLR 2023
> > > >
> > > > [D] Deep Graph Infomax, ICLR 2019
> > > >
> > > > [E] Contrastive Multi-View Representation Learning on Graphs, ICML 2020
> > > >
> > > > [F] Deep Graph Contrastive Representation Learning, ICML Workshop on Graph Representation Learning and Beyond 2020
> > > >
> > > > [G] Simple Unsupervised Graph Representation Learning, AAAI 2022

---

> > > > > ### Comment · Reviewer_Ck44 · 2023-08-17
> > > > > **Re: Re: Baselines**
> > > > >
> > > > > Thanks for compiling the table, that is very useful.
> > > > >
> > > > > However, [A-C] also have these reference results in _comparable_ settings, right? How do these compare to your reproduced numbers?
> > > > >
> > > > > Just to pick an example, Geom-GCN-I (Table 3, [A]) obtains  90.05 % on Pubmed, vs. 84.18-88.60% in your Table 1.
> > > > >
> > > > > Could you please expand on how the numbers reported in A-C compare to your reproduced methods, and how they compare to FIGURe?

---

> > > > > > ### Author Response · Authors · 2023-08-19
> > > > > >
> > > > > > We appreciate the reviewer's feedback.
> > > > > >
> > > > > > We wish to emphasize that our approach falls under the unsupervised paradigm, focusing on learning representations for individual filters without supervision. The subsequent acquisition of combination coefficients is tailored to the specific task at hand. It's worth noting that unlike our method, GPRGNN [A] and H2GCN [B] adopt a supervised stance, where GPRGNN simultaneously handles both representation and combination coefficient learning. Our work illustrates the potential of disentangling the representation learning process from that of combination coefficient acquisition. Our supplementary material, specifically Table-9, provides a comparative analysis involving state-of-the-art (SOTA) models such as GPRGNN and H2GCN, recognized for their efficacy across both homophilic and heterophilic datasets. This table is provided here for your convenience. We would like to point out that GPRGNN and H2GCN do better (on an average) than most of the other baselines such as GeomGCN [C], FAGCN [D], LGC [E], SuperGAT [F] etc. The results have been borrowed from [G].
> > > > > >
> > > > > > Our selection of GPRGNN was based on its recognized status as a state-of-the-art (SOTA) model and its strong alignment with our approach. Specifically, our model follows an unsupervised route, first learning representations and then combining them based on the downstream task, while GPRGNN takes on both tasks simultaneously. Similarly, we considered H2GCN due to its innovative approach in addressing heterophily by isolating node and neighbor embeddings. Moreover, this model aggregated embeddings from K-different hops and then concatenates them all. This is similar to both the GPRGNN model, and our model. We would like to point out that we don't compare against PPGNN [G], as PPGNN requires eigendecomposition, which could be computationally expensive for larger graphs.
> > > > > >
> > > > > > Our findings illustrate that, despite operating within the unsupervised framework, our approach outperforms these SOTA methods on 4 out of the 8 datasets we assessed. Furthermore, on the remaining datasets, our performance closely approximates theirs. We hold the perspective that with continued refinement, we have the potential to surpass the performance of these SOTA supervised methods as well.
> > > > > >
> > > > > > In addition, we'd like to highlight that our approach exhibits superior performance among various unsupervised learning methods, including those considered state-of-the-art (SOTA) in Table 1.
> > > > > >
> > > > > >
> > > > > > |           | SQUIRREL     | CHAMELEON    | ROMAN-EMPIRE | MINESWEEPER  | ARXIV-YEAR   | OGBN-ARXIV   | CORA         | CITESEER     | PUBMED       |
> > > > > > |-----------|--------------|--------------|--------------|--------------|--------------|--------------|--------------|--------------|--------------|
> > > > > > | GCN       | 47.78 (2.13) | 62.83 (1.52) | 73.69 (0.74) | 89.75 (0.52) | 46.02 (0.26) | 69.37 (0.00) | 87.36 (0.91) | 76.47 (1.34) | 88.41 (0.46) |
> > > > > > | GPRGNN    | 46.31 (2.46) | 62.59 (2.04) | 64.85 (0.27) | 86.24 (0.61) | 45.07 (0.21) | 68.44 (0.00) | 87.77 (1.31) | 76.84 (1.69) | 89.08 (0.39) |
> > > > > > | H2 GCN    | 37.90 (2.02) | 58.40 (2.77) | 60.11 (0.52) | 89.71 (0.31) | 49.09 (0.10) | OOM          | 87.81 (1.35) | 77.07 (1.64) | 89.59 (0.33) |
> > > > > > | FiGURe$_{32}$  | 48.89 (1.55) | 65.66 (2.52) | 67.67 (0.77) | 85.28 (0.71) | 41.30 (0.21) | 66.58 (0.00) | 82.56 (0.87) | 71.25 (2.20) | 84.18 (0.53) |
> > > > > > | FiGURe$_{128}$ | 48.78 (2.48) | 66.03 (2.19) | 68.10 (1.09) | 85.16 (0.58) | 41.94 (0.15) | 69.11 (0.00) | 86.14 (1.13) | 73.34 (1.91) | 85.41 (0.52) |
> > > > > > | FiGURe$_{512}$    | 52.23 (1.19) | 68.55 (1.87) | 70.99 (0.52)  | 85.58 (0.49) | 42.26 (0.20) | 69.69 (0.00) | 87.00 (1.24) | 74.77 (2.00) | 88.60 (0.44)  |
> > > > > >
> > > > > >
> > > > > > ---
> > > > > >
> > > > > > ## References
> > > > > >
> > > > > > [A] Chien, Eli, et al. Adaptive universal generalized pagerank graph neural network. ICLR 2021.
> > > > > >
> > > > > > [B] Zhu, Jiong, et al. Beyond homophily in graph neural networks: Current limitations and effective designs. NeurIPs, 2020.
> > > > > >
> > > > > > [C] Hongbin Pei, Bingzhe Wei, Kevin Chen-Chuan Chang, Yu Lei, and Bo Yang. Geom-gcn: Geometric graph convolutional networks. ICLR, 2020
> > > > > >
> > > > > > [D] Deyu Bo, X. Wang, Chuan Shi, and Hua-Wei Shen. Beyond low-frequency information in graph convolutional networks. AAAI, 2021
> > > > > >
> > > > > > [E] Johannes Klicpera, Aleksandar Bojchevski, and Stephan Gunnemann. Combining neural networks with personalized pagerank for classification on graphs. In International Conference on Learning Representations ICLR, 2019
> > > > > >
> > > > > > [F] Dongkwan Kim and Alice Oh. How to find your friendly neighborhood: Graph attention design with self-supervision.  ICLR, 2021.
> > > > > >
> > > > > > [G] A Piece-wise Polynomial Filtering Approach for Graph Neural Networks, ECML PKDD 2022

---

> > > > > > > ### Comment · Reviewer_Ck44 · 2023-08-19
> > > > > > > **Updated score to weak accept**
> > > > > > >
> > > > > > > Dear authors,
> > > > > > >
> > > > > > > thanks a lot for the additional discussion. I also appreciate the summary you posted with updates to the paper.
> > > > > > >
> > > > > > > To reflect the additional insights I got through the rebuttal period, I updated my score to a 6 / weak accept. I share some of the other reviewer's concerns that the method might still be too incremental (despite your additional clarification regarding the contribution), hence I did not increase further. Yet, I now find your experimental results more convincing, and think that they add to the existing literature. I should note that this paper is out of my primary field, hence I leave my confidence for the rating at 3.
> > > > > > >
> > > > > > > Regarding my concern,
> > > > > > >
> > > > > > > > Explaining Augmentation: We acknowledge the inconsistency in the usage of the term "augmentation" and will strive to enhance clarity by using the term consistently and providing a more detailed explanation in Section 5.
> > > > > > >
> > > > > > > it would be great if you could also share the respective updated paragraph from the paper as a comment here.

---

> > > > > > > > ### Author Response · Authors · 2023-08-21
> > > > > > > >
> > > > > > > > We extend our sincere gratitude to the reviewer for their invaluable contribution to the rebuttal process and for offering insightful comments that have significantly enriched our work. We also appreciate the reviewer's recognition of the merits of our research, reflected in the increased score. We have reproduced the paragraph (below) present in line numbers 134 - 141 in the updated version of the paper. The updated manuscript is accessible through the anonymized link provided in the supplementary material at Line 471, Section 8.1.
> > > > > > > >
> > > > > > > >
> > > > > > > > **Reproduced Paragraph:**
> > > > > > > >
> > > > > > > > Our method FiGURe~(**Fi**lter-based **G**raph **U**nsupervised **Re**presentation Learning)  builds on concepts introduced in [A, B], extending the maximization of mutual information between node and global filter representations for each filter in the filter bank $F$ = {$F_1$, $F_2$, ... $F_K$}. In this approach, we employ filter-based augmentations, treating filter banks as "additional views" within the context of contrastive learning schemes. In the traditional approach, alternative baselines like DGI have employed a single filter in the GPRGNN [C] filter bank. Additionally, MVGRL [D] attempted to use the diffusion kernel; nevertheless, they only learned a single representation per node. We believe that this approach is insufficient for accommodating a wide range of downstream tasks. We construct an encoder for each filter to maximize the mutual information between the input data and encoder output.
> > > > > > > >
> > > > > > > >
> > > > > > > > ---
> > > > > > > > **References**
> > > > > > > >
> > > > > > > > [A] R. D. Hjelm, A. Fedorov, S. Lavoie-Marchildon, K. Grewal, P. Bachman, A. Trischler, and Y. Bengio. Learning deep representations by mutual  information estimation and maximization. ICLR, 2019.
> > > > > > > >
> > > > > > > > [B] P. Velickovic, W. Fedus, W. L. Hamilton, P. Liò, Y. Bengio, and R. D. Hjelm. Deep Graph Infomax. ICLR, 2019
> > > > > > > >
> > > > > > > > [C] Chien, Eli, et al. Adaptive universal generalized pagerank graph neural network. ICLR 2021.
> > > > > > > >
> > > > > > > > [D] Contrastive Multi-View Representation Learning on Graphs, ICML 2020

---

### Official Review · Reviewer_qpib · 2023-07-24

**Soundness:** 4 excellent
**Presentation:** 4 excellent
**Contribution:** 3 good
**Rating:** 6
**Confidence:** 3

**Summary:**

The paper proposes a few approaches to improve the contrastive learning framework of the unsupervised graph representation learning (UGRL) problem. Building on top of the prior works in supervised GRL and UGRL areas (e.g., filter bank construction, etc.), the authors argued that 1) filter-based augmentations (essentially treating filter banks as "additional views" in contrastive learning schemes) is able to provide useful representations across cases that require high-freq and low-freq components; and 2) we can leverage a lot of techniques long known in the ML/DL community like RFF to efficiently reduce the computational complexity of these latent representations in UGRL.

**Strengths:**

The paper proposed a relatively novel approach of leveraging different filter banks as additional views in a contrastive graph representation learning problem. The method basically builds upon the MI maximization scheme proposed in methods like DGI but extends the approach to also include filter bank $F=\{F_1, \dots, F_k\}$ such that each filter's encoder output's MI with the input data is maximized. The empirical analysis reveals that the approach is superior (and likely compatible with?) to the previous UGRL approaches that didn't use filter-bank-related augmentations; and that RFF provides a reasonable boost to low-dimensionality representation learning in graphs. Specifically:

1. The paper provides a reasonable empirical analysis of the approach on a diverse set of heterophilic and homophilic datasets, demonstrating improvements in both settings.
2. Numerous ablative studies were made, including on low and high dimensionalities, encoder weight sharing, efficiency, and filter bank selection.
3. The idea of including multiple filters in the contrastive learning framework itself makes sense to me and is a novel addition to the UGRL literature.

**Weaknesses:**

A few weaknesses of the paper:

1. The main idea of the approach (maximizing mutual information of encoder output and input; applying discriminator to the patch-readout pair), which is described in Sec. 5.1, is still mainly built up on the discussion of the DGI paper. Although the inclusion of the filter bank discussion is new (which follows from other recent work like BernNet, GPRGNN, and ChebNet), this still limits the novelty of the approach itself.
2. The contribution of RFF projections, as shown in Sec. 6.2, is largely orthogonal to the FiGURe approach itself. I wonder whether it's the best idea to include it in this paper whose title is on FiGURe and filter augmentations. For example, the efficiency analysis in Sec. 6.4 clearly is lacking the comparison to DGI/MVGRL + RFF, and is thus unfair. Moreover, as the authors suggested in the supplementary materials, the RFF projection they used was based on a Gaussian kernel. Whether and how different projections (e.g., Laplacian) make a difference is unknown; the gradient instability of using RFFs in the architecture (which is an important problem in RFFs' usage in Transformers) is not discussed, etc. I personally feel there is a lot of value to the analysis of RFF itself.
3. I'm less certain whether "shared encoder weight" is a contribution (as listed on line 48) of this paper. It seems more like a remedy to the computational burden added by the FiGURe approach, but is not required otherwise?

**Questions:**

1. What is the effect of the number of encoder layers used in this case? The authors used one (as was in DGI), but there have also been works that used more.
2. Could the authors expand Table 5 to include RFF + {other baseline methods}?
3. Are the numbers provided in Table 1 and 2 the results of the authors' reproduction, or the original papers? Some of these numbers were generally different from what was reported in the original & other papers (e.g., MVGRL, GRACE).
4. Could the authors elaborate more on the corruption function $C$ (especially how it creates $\tilde{F}$)?

---------

UPDATE: I have read the rebuttals from the authors and appreciate the additional experiments & explanations. As the authors acknowledge, the benefit of RFF is not unique to FiGUREe (which some other reviewers also pointed out), and I think it should be elaborated on and compared to more clearly in the paper's revision (so is the novelty issue). I'd like to maintain the current rating, but I think the paper's writing can be improved more on these axis to avoid confusion.

---

> ### Author Rebuttal · Authors · 2023-08-09
>
> Firstly, we would like to extend our sincere gratitude for your comprehensive and constructive feedback on our paper. We appreciate your recognition of the strengths in our work. Specifically, we are pleased that you found value in our novel approach of leveraging different filter banks as additional views in the contrastive graph representation learning problem. Your acknowledgment of the benefits of this method, including its empirical analysis across a diverse set of heterophilic and homophilic datasets, and its potential novelty addition to the literature is encouraging.
>
> # Comment Response
> 1. **Novelty:** We acknowledge that our approach in Section 5.1 is largely built upon the foundational ideas from the DGI paper and that prior works have incorporated various filter banks. However, prior filter bank methods jointly learn the filter specific representations and the combination coefficients. The novel insight that we are proposing in this paper is that we can separately learn these two parts and **bridge the gap with supervised methods**. This enabled us to **propose a simple extension** to DGI which easily incorporates these filter banks. Extending other methods to use filter banks is non-trivial and we plan to explore these in our future work. We will make these aspects more clear in our paper.
> 2. **Inclusion of RFF:** Our goal was to ensure that augmenting these additional filter banks does not make the approach impractical by significantly increasing the training time. In Section 6.4, we wished to demonstrate that with RFF we are able to make the training time practical enough. Your feedback on the inclusion of RFF projections and its relation to the FiGURe approach is noted. We realize the potential for separate deep dives into the contributions of RFF and FiGURe and will consider refining our content accordingly. Also, here is a table with performance numbers with few different projections.
> | Features                    | Cora         | Citeseer     | Squirrel     | Chameleon    |
> |-----------------------------|--------------|--------------|--------------|--------------|
> | Polynomial (d=2) [A]  | 81.35 (2.11) | 69.81 (2.14) | 38.57 (1.56) | 53.88 (1.94) |
> | Polynomial (d=10) [A] | 80.44 (1.56) | 68.71 (1.67) | 38.08 (1.10) | 55.20 (1.97) |
> | Exp [A]             | 80.42 (2.14) | 68.81 (1.34) | 38.06 (1.61) | 55.50 (2.11) |
> | ANOVA [B]                | 83.26 (0.78) | 70.09 (2.44) | 40.77 (1.46) | 56.01 (1.86) |
> | RFF | 87.00 (1.24) | 74.77 (2.00) | 52.23 (1.19) | 68.55 (1.87) |
>
> We will add more details about the training setup and results on all dataset to the paper as suggested. We will also add the discussion on the gradient instability of using RFFs in the architecture.
>
> 3. **Effect of Encoder Layers:** Refer to global response.
> 4. **Numerical Differences:** Refer to global response.
> 5. **Clarity of Corruption Function:** Refer to global response.
>
> * [A] Random Feature Maps for Dot Product Kernels. AISTATS 2012.
> * [B] Random Feature Maps for the Itemset Kernel. AAAI 2019.

---

### Author Rebuttal · Authors · 2023-08-10

We sincerely thank the reviewers for their valuable feedback. Reviewer qpib's recognition of our novel approach leveraging filter banks is appreciated. Reviewer Ck44's positive remarks on clear presentation and readable figures are noted. Reviewer wrwF's acknowledgment of our approach's versatility and clarity in contributions is encouraging. Reviewer xu4A's appreciation of our paper's quality, consistent performance of FIGURe, and improvement over other methods highlights the strength of our work.

We have addressed specific concerns that the reviewers have raised as responses to the individual reviews. Here we address some of the common concerns that were raised by two or more reviewers.

1. **Equation discrepancy**: Equation 3 has been fixed: $\mathcal{L}\_{F\_i} = -\frac{1}{N + M}\left( \sum\_{j=1}^{N}\mathbb{E}[\text{log}(D\_{\omega}(h\_{j}^{F\_{i}}, h\_{g}^{F\_{i}}))] + \sum\_{j=1}^{M}\mathbb{E}[\text{log}(1 - D\_{\omega}(\widetilde{h\_{j}^{F\_{i}}}, h\_{g}^{F\_{i}}))] \right)$

2. **Numerical differences:** There are some differences in the numbers reported in the original papers of the baseline methods and our reproduction. The dataset splits used are different from the ones used in the original papers. Specifically, the splits we use have been used before in papers like [A,B,C]. The reason we chose these splits is two-fold. Firstly, there are ten splits provided in these papers (except for the large datasets), as opposed to the single split versions that the baseline methods are working with. We report all our results as averages over all these 10 splits. Secondly, there are certain inconsistencies in some of the splits being used in the baselines. For example, for cora, the dataset split used by [D] is different from the one used utilised in the original [E] paper, which is the split they use for reporting their baseline performance numbers. For more details on this, issue \#2 on the MVGRL GitHub repository can be referred to. For all the baselines, we ran the code provided by the authors of those works along with extensive hyperparameter tuning. Specifically, we separate the hyperparameter tuning of the unsupervised part and supervised part, and do not use any information from the supervised part while tuning the hyperparameters of the unsupervised part.


3. **Encoder layers:** We include an analysis with more encoder layers. There are two interpretations of 'encoder layers' in this case. First, a deeper GCN, which implies the aggregation of multiple hop neighborhood information into the node. Second, a single hop GCN with a deeper network to transform the features. We report the performance of both these cases, with two and three layers each. It can be observed that the single layer GCN performs equal or better than all other cases.

| Accuracies            | Cora | Citeseer |  Squirrel | Chameleon |
|-------------|------------------|---------------------|---------------------|----------------------|
| 1 Layer GCN | 87.00 (1.24)     | 74.77 (2.00)        | 52.23 (1.19)        | 68.55 (1.87)         |
| 2 Layer GCN | 86.62 (1.43)     | 73.62 (1.46)        | 43.80 (1.57)        | 53.53 (2.13)         |
| 3 Layer GCN | 84.40 (1.84)     | 72.52 (2.09)        | 42.79 (1.12)        | 61.73 (2.25)         |
| GCN + 2 Layer MLP | 85.73 (1.03)     | 70.21 (2.30)        | 49.91 (2.68)        | 68.18 (1.76)         |
| GCN + 3 Layer MLP | 84.99 (1.43)     | 71.39 (2.32)        | 45.85 (3.33)        | 64.19 (1.43)         |


4. **Clarity of Corruption Function**: The corruption function takes nodes and their associated features ($\mathbf{X}$) as input, along with a graph ($\mathbf{F}_{i}$) representing these nodes. The function maintains the structure of the graph, but it rearranges the rows of $\mathbf{X}$. This rearrangement effectively shuffles the node features, ensuring that each node's features differ from the original input data. When generating embeddings using a Graph Convolutional Network (GCN), the node embeddings are combined, resulting in a corrupted representation of the nodes.


* [A] Geom-GCN: Geometric Graph Convolutional Networks, ICLR 2020
* [B] A Piece-wise Polynomial Filtering Approach for Graph Neural Networks, ECML PKDD 2022
* [C] A critical look at the evaluation of GNNs under heterophily: are we really making progress? ICLR 2023
* [D] Contrastive Multi-View Representation Learning on Graphs. ICML 2020
* [E] Semi-Supervised Classification with Graph Convolutional Networks. ICLR 2017

---

### Author Response · Authors · 2023-08-19
**Rebuttal Summary**

We extend our gratitude to the reviewers for their constructive critiques and invaluable feedback. Throughout the rebuttal phase, we have taken the following steps to address the concerns raised:

We have revised certain parts of Section 5.1. The updated manuscript is accessible through the anonymized link provided in the supplementary material at Line 471, Section 8.1. For the final camera-ready version, we will ensure that the main paper fits within the required page limits.

We have incorporated all the additional experiments discussed during the rebuttal into the supplementary material:

a. In-depth analysis with an increased number of encoder layers.

b. Exploration of alternative random projection techniques.

c. Examination of a distinct filter bank derived from Chebyshev Polynomials.

d. In-depth analysis of the alpha coefficients (combination coefficients of the GPRGNN filter).

e. Assessment of RFF's efficacy even at 512 dimensions.

f. Performance evaluation of other baseline methods with RFF at 128 dimensions.

g. Comprehensive reporting of time and storage costs across various embedding sizes, alongside a comparison with DGI.

We have introduced additional insights within the main paper and further elucidated experimental specifics within the supplementary material.

As the rebuttal phase approaches its conclusion, we invite the reviewers to reach out with any remaining queries or clarifications. We hold the belief that our extensive experiments and elucidations have addressed their concerns, strengthening the robustness of our paper. We kindly request their favorable consideration of our submission.

---

### Decision · Program_Chairs · 2023-09-21

**Decision:**

Accept (poster)

**Comment:**

Paper Summary

The paper proposes an approach to learning representations of nodes using contrastive learning. The authors show a new augmentation scheme that is particularly effective and also show a low-dimensional feature mapping scheme using Random Fourier Features that improves performance.


Reviews & Justification

The paper received mostly positive reviews. One reviewer raised concerns about code and notational issues which the authors have addressed satisfactorily. Multiple reviewers raised concerns about the applicability of the work to multi-layer GNNs. The authors have addressed this concern in their response. The AC encourages the authors to add these experiments to the final paper as well. Finally, the authors also provided an in-depth experimental analysis of RFF dimensions, the alpha coefficients, and examined a different filter bank.